# Do MLPs Inherit GNN Knowledge Uniformly? Class-Conditional Transfer Equity in Graph-Free Distillation

## Abstract

Graph Neural Networks (GNNs) achieve strong performance by leveraging relational structure, but their reliance on message passing limits deployment in latency-sensitive settings. Distilling GNNs into Multilayer Perceptrons (MLPs) provides a feature-only student that approximates a graph-aware teacher without message passing at inference. Existing studies typically evaluate such transfer using average accuracy, implicitly assuming that teacher knowledge is inherited uniformly across labels. However, an MLP may match the teacher well on average while failing to inherit confident teacher knowledge for particular classes. We study this failure mode as class-conditional missed transfer in graph-free distillation. We first introduce *Valid Transfer Agreement*, a label-level metric that measures whether a graph-free student preserves teacher decisions on test samples where the teacher is correct. To mitigate this failure mode, we propose `TED` (Transfer Equity Distillation), a training-time objective augmentation for existing GNN-to-MLP distillation methods. `TED` formulates reliability-guided distillation by treating correct and confident teacher decisions as transferable knowledge and defining label-wise transfer hardness as the portion of this knowledge not inherited by the student. By optimizing a differentiable hardness-based objective, `TED` reduces excessive missed transfer for difficult labels while leaving the teacher, student architecture, and inference procedure unchanged. Across eight homophilic and heterophilic graph benchmarks, `TED` reduces class-level transfer disparity and improves worst-label teacher–student agreement while preserving competitive accuracy and graph-free inference efficiency. Source code is available at: TED.

## 1 Introduction

Graph-structured data naturally models relational dependencies in many modern applications, making representation learning on graphs a central problem in machine learning (Li et al., 2022; Barman et al., 2026; Wang et al., 2021; Wu et al., 2022). Graph Neural Networks (GNNs) have become a dominant paradigm for learning on such data. By propagating and aggregating information over graph structure, GNNs exploit both node attributes and connectivity, often outperforming feature-only models on node classification and related tasks (Kipf & Welling, 2016; Hamilton et al., 2017; Veličković et al., 2017). This representational advantage, however, comes with a substantial deployment cost. A single GNN prediction may require recursive neighborhood expansion, feature fetching, and message aggregation, making inference expensive on large graphs and difficult to serve in latency-sensitive systems (Jia et al., 2020; Min et al., 2021). Multilayer Perceptrons (MLPs) provide an appealing alternative, offering a simple, fast, and graph-free inference model. Yet, because they operate only on node features, they lack the relational inductive bias that makes GNNs effective for structured data.

Knowledge distillation offers a practical route for reconciling this trade-off between graph-aware expressivity and graph-free deployment (Hinton et al., 2015; Li et al., 2025). In GNN-to-MLP distillation, a topology-aware GNN teacher is first trained using graph structure, and its predictive knowledge is then transferred to an MLP student that uses only node features at test time (Zhang et al., 2021). Prior work has shown that such students can substantially improve over vanilla MLPs while removing inference-time graph dependency. Subsequent methods further strengthen this paradigm by injecting structural signals, refining distillation

Figure 1: **Class-conditional transfer disparity in graph-free GNN-to-MLP distillation.** We report class-wise Valid Transfer Agreement (VTA) on CiteSeer and Amazon-Photo under both transductive and inductive settings. Although the base graph-free student achieves competitive average accuracy, VTA reveals substantial variation in how correctly predicted teacher decisions are inherited across labels. The `TED`-augmented student reduces this disparity by improving transfer for under-transferred classes, yielding a lower Equity Gap while preserving competitive predictive accuracy. Furthermore, `TED` also improves the VTA of the most under-transferred class under vanilla GLNN.

objectives, and analyzing the nature of transferable graph knowledge (Tian et al., 2022; Wu et al., 2023; Singh et al., 2026; 2025). Together, these approaches have significantly advanced graph-free learning from GNN teachers.

Despite this progress, existing evaluations of GNN-to-MLP distillation remain dominated by average accuracy. Such aggregate metrics can obscure whether the student inherits teacher knowledge uniformly across labels. This is a consequential omission because graph-free distillation changes the information available at inference time. The GNN teacher predicts using both node features and graph structure, whereas the deployed MLP student must approximate the teacher using node features alone. This loss of structural access need not affect all labels equally. Some classes may be largely recoverable from node attributes, while others may depend more strongly on neighborhood context, structural position, high-frequency graph signals, or heterophilic evidence. As a result, an MLP can match the teacher well on average while failing to preserve valid teacher knowledge for particular classes.

In this paper, we study this phenomenon as *class-conditional transfer disparity*. Importantly, transfer disparity refers to the *outcome* of distillation, not to identical treatment of labels during optimization. Our concern is whether the final graph-free student inherits valid GNN teacher knowledge uniformly across labels after training. To make this notion measurable, we consider teacher decisions that are correct and ask whether the student preserves them label by label. We call this label-wise agreement with correct teacher decisions *Valid Transfer Agreement* (VTA), and later formalize it in Section 3. A large spread in VTA across labels indicates that some classes lose substantially more valid teacher knowledge than others.

Figure 1 illustrates this failure mode on CiteSeer and Amazon-Photo under both transductive and inductive settings. Some classes exhibit substantially lower VTA, meaning that the MLP fails to preserve correct teacher decisions for those labels despite strong aggregate performance. This suggests that graph-free deployment can introduce a label-dependent loss of GNN knowledge that is invisible to average accuracy.

**Research question.** This paper asks: *Do graph-free MLP students inherit reliable GNN teacher knowledge uniformly across labels, and can distillation be constrained to reduce class-conditional transfer disparity without sacrificing predictive utility?*

We answer this question by using VTA as the evaluation lens and designing `TED` (***T**ransfer **E**quity*[1] ***D**istillation*), a training-time objective augmentation for graph-free GNN-to-MLP distillation. For each class, VTA considers the samples on which the GNN teacher predicts correctly and measures how often the MLP student predicts the same label as the teacher. By conditioning on teacher correctness, VTA separates missed transfer from cases where the teacher itself is wrong, enabling a more precise diagnosis of whether

---

[1]In this work, **"equity" refers only to *class-conditional transfer disparity*** in graph-free GNN-to-MLP distillation. It should not be interpreted as demographic fairness. We do not treat class labels as sensitive attributes, nor do we claim fairness guarantees for protected groups or downstream populations. Our focus is a deployment reliability problem: a graph-free student should not systematically lose valid GNN teacher knowledge for a subset of classes.

valid teacher knowledge is inherited uniformly across labels. We summarize this class-conditional disparity using the Equity Gap, defined as the range between the highest and lowest class-wise VTA values.

The key challenge is to reduce this disparity without forcing the student to imitate unreliable teacher outputs. If the teacher is incorrect or uncertain, enforcing agreement may propagate noise rather than useful graph knowledge, thereby degrading predictive utility. We therefore treat teacher decisions as reliable only when they are both correct and confident. Building on this reliability view, `TED` defines the transferable hardness of each label as the amount of reliable teacher knowledge that the student fails to inherit. `TED` penalizes labels whose transferable hardness exceeds the average by more than a tolerance budget.

`TED` augments existing GNN-to-MLP distillation objectives with a class-level penalty that targets disproportionate missed transfer across labels. Since the constraint is used only during training, `TED` does not alter the student architecture, inference procedure, or graph-free deployment setting. Experiments across graph benchmarks show that `TED` reduces class-level transfer gaps and improves worst-label teacher–student agreement while maintaining competitive average accuracy.

Our contributions are summarized as follows:
❶ We identify *class-conditional transfer disparity* as an overlooked reliability failure in graph-free GNN-to-MLP distillation.
❷ We introduce *Valid Transfer Agreement* (VTA), a label-level metric that measures how faithfully the student preserves correct teacher decisions.
❸ We formulate class-conditional transfer reliability as a bounded excess-risk problem. Specifically, we define a teacher-conditioned missed-transfer risk for each class and require that no class exceed the average transfer risk by more than a tolerance budget.
❹ We propose `TED` (Transfer Equity Distillation), a plug-in training-time augmentation for existing graph-free distillation methods. Starting from a bounded class-conditional excess-risk constraint, we derive `TED` as a selective squared-hinge regularizer that penalizes only classes exhibiting excessive missed transfer of reliable teacher knowledge. `TED` reduces class-level transfer disparity and improves worst-label teacher–student agreement while preserving the student architecture, graph-free inference, and competitive average accuracy.

## 2 Related Works

GNN-to-MLP knowledge distillation has emerged as a practical approach for deploying graph models without inference-time message passing. Graph-less Neural Networks (GLNN) (Zhang et al., 2021) established this paradigm by training an MLP student with soft-label supervision from a GNN teacher, allowing the student to use only node features at inference. Subsequent methods improve MLP students by exposing them to additional structural signals. NOSMOG (Tian et al., 2022) augments node attributes with DeepWalk-based positional embeddings (Perozzi et al., 2014), SA-MLP (Chen et al., 2024) encodes the adjacency matrix through a linear layer, and VQGraph (Yang et al., 2024) learns a structure-aware tokenizer over local graph patterns to construct richer distillation targets. While effective, these methods rely on graph-derived representations or preprocessing during inference, and therefore differ from strictly graph-free deployment. We focus on the stricter regime where the deployed student consumes only the original node attributes. Class-view graph distillation changes the distillation unit from sample-wise prediction matching to class-wise prediction matching to improve accuracy, whereas our work measures and regularizes whether correct teacher decisions are inherited uniformly across classes (Tian et al., 2025).

More recently, trustworthiness in GNN-to-MLP distillation has been studied (Capetz et al., 2025). FAITH (Singh et al., 2026) improves group and individual fairness with respect to sensitive attributes during graph-free distillation. Furthermore, few methods have used the reliability of GNN for distillation. KRD (Wu et al., 2023), HGMD (Wu et al., 2024), and InfGraND (Eskandari et al., 2026) use reliability and hardness estimates in GNN-to-MLP distillation. KRD measures the stability of teacher predictions under perturbations and emphasizes reliable nodes during training. HGMD uses entropy-based knowledge hardness and hardness-aware subgraphs to supervise difficult samples. InfGraND uses graph influence to identify structurally important nodes for stronger distillation. Their objective is to improve the fidelity of instance-level distillation signals. These approaches aim to improve the informativeness of the distillation signal at the instance or subgraph level. Our use of reliability is conceptually different. Rather than ask-

ing which samples should receive stronger supervision, we use reliability to define which teacher decisions constitute valid transferable knowledge, and then control how this knowledge is inherited across classes.

Beyond GNN-to-MLP distillation, class imbalance and worst-group performance have been widely studied through reweighting and robust optimization objectives. Class-balanced losses reweight training samples according to class frequency or effective sample number to compensate for label imbalance (Zhang et al., 2023). Focal loss emphasizes examples that are difficult under the current student prediction, thereby reducing the dominance of easy examples during training (Lin et al., 2017; Tan et al., 2022). Distributionally robust optimization (DRO) minimizes a worst-group or worst-class risk and has been used to improve robustness under group imbalance or distributional shifts (Wang et al., 2023; Vilouras et al., 2023). Class-balanced loss uses label frequency, focal loss uses student-side prediction hardness, and DRO uses average group loss. TED instead defines a class-wise missed reliable transfer residual using teacher-correct and confident predictions, and regularizes this transfer-specific residual to reduce disparities in valid teacher-knowledge inheritance. Thus, TED is aligned with VTA, whereas reweighting and robust optimization methods do not explicitly distinguish between ordinary classification difficulty and failure to inherit correct teacher decisions.

To the best of our knowledge, prior GNN-to-MLP distillation methods have not explicitly examined whether reliable teacher knowledge is inherited uniformly across labels. Our work fills this gap by introducing a label-level transfer metric and a training-time objective that controls class-conditional missed transfer while preserving graph-free inference.

## 3 Preliminaries and Problem Formulation

Let $\mathcal{G} = (\mathcal{V}, \mathcal{E})$ be a graph with $N = |\mathcal{V}|$ nodes, node-feature matrix $\mathbf{X} \in \mathbb{R}^{N \times d}$, and adjacency matrix $\mathbf{A} \in \{0, 1\}^{N \times N}$. We denote the feature vector of node $v$ by $\mathbf{x}_v$. Each node has a label $y_v \in \{1, \ldots, C\}$, and $\mathcal{V}_L \subseteq \mathcal{V}$ denotes the labeled node set used for training and evaluation. For class $c$, let $\mathcal{V}_c = \{v \in \mathcal{V}_L : y_v = c\}$ denote the labeled nodes of class $c$. We study node classification under GNN-to-MLP distillation, where a graph-aware teacher is used to supervise a graph-free student.

**Teacher GNN.** The teacher is a pre-trained and frozen GNN $f_\theta$. For each node $v$, the teacher predicts using both node features and graph structure:

$$\mathbf{p}_v^t = f_\theta(\mathbf{X}, \mathbf{A})_v \in \Delta^{C-1}, \qquad \hat{y}_v^t = \arg\max_{k \in \{1, \ldots, C\}} \mathbf{p}_{v,k}^t. \tag{1}$$

Here, $\Delta^{C-1}$ denotes the probability simplex over $C$ classes, and $\mathbf{p}_{v,k}^t$ denotes the teacher-assigned probability that node $v$ belongs to class $k$. Because $f_\theta$ performs message passing over $\mathbf{A}$, its prediction may use neighborhood information, structural position, and other relational signals that are unavailable to a feature-only model.

**Student MLP.** The student is an MLP $g_\phi$ that receives only the node feature $\mathbf{x}_v$:

$$\mathbf{p}_v^s = g_\phi(\mathbf{x}_v) \in \Delta^{C-1}, \qquad \hat{y}_v^s = \arg\max_{k \in \{1, \ldots, C\}} \mathbf{p}_{v,k}^s. \tag{2}$$

At inference time, the student does not access $\mathbf{A}$, node neighborhoods, structural encodings, or the teacher. The deployed predictor is therefore strictly graph-free and depends only on the original node attributes.

**Valid teacher knowledge.** Our goal is not to force the student to imitate every teacher prediction. A teacher prediction is useful for evaluating transfer only when it is correct. We therefore define the set of valid teacher decisions for class $c$ as

$$\mathcal{T}_c = \{v \in \mathcal{V}_c : \hat{y}_v^t = y_v\}. \tag{3}$$

These are the labelled nodes of class $c$ on which the teacher predicts the ground-truth label. Conditioning on $\mathcal{T}_c$ allows us to separate missed transfer from cases where the teacher itself is incorrect.

### 3.1 Class-Conditional Transfer Metrics

Average accuracy and aggregate teacher–student agreement do not reveal whether teacher knowledge is inherited uniformly across labels. We therefore evaluate GNN-to-MLP distillation at the class level, focusing on teacher decisions that are correct and hence valid for measuring knowledge transfer.

For each class $c$, let $\mathcal{T}_c = \{v \in \mathcal{V}_c : \hat{y}_v^t = y_v\}$ denote the set of test nodes on which the teacher predicts correctly, and let $\mathcal{C}_{\text{valid}} = \{c : |\mathcal{T}_c| > 0\}$ denote the set of classes with at least one correct teacher decision. Classes outside $\mathcal{C}_{\text{valid}}$ are excluded from class-level transfer analysis.

**Definition 1** (Valid Transfer Agreement). *For class* $c \in \mathcal{C}_{\text{valid}}$, *the* Valid Transfer Agreement *is*

$$\text{VTA}_c = \frac{1}{|\mathcal{T}_c|} \sum_{v \in \mathcal{T}_c} \mathbf{1}[\hat{y}_v^s = \hat{y}_v^t]. \tag{4}$$

*It measures the fraction of correct teacher decisions for class c that are preserved by the graph-free student.*

**Definition 2** (Equity Gap). *The* Equity Gap *summarizes class-conditional transfer disparity as*

$$\text{EG} = \max_{c \in \mathcal{C}_{\text{valid}}} \text{VTA}_c - \min_{c \in \mathcal{C}_{\text{valid}}} \text{VTA}_c. \tag{5}$$

*A smaller* EG *indicates a smaller spread in class-wise VTA values.*

### 3.2 Problem Formulation

We study equitable graph-free knowledge transfer. Given a graph $\mathcal{G} = (\mathcal{V}, \mathcal{E})$, node features $\mathbf{X}$, labels on $\mathcal{V}_L$, a pre-trained frozen GNN teacher $f_\theta$, and an MLP student $g_\phi$, our goal is to train a feature-only student that preserves the deployment efficiency of GNN-to-MLP distillation while inheriting valid teacher knowledge more uniformly across classes.

This objective has two desiderata. First, the graph-free student should preserve **predictive utility**, achieving competitive average node-classification performance. Second, it should improve **class-conditional transfer reliability**, reducing disparities in the inheritance of valid teacher knowledge as measured by the Equity Gap in Eq. (5).

## 4 Methodology

We propose `TED`, a training-time objective augmentation for graph-free GNN-to-MLP distillation. Given an existing distillation method, `TED` keeps the teacher, student architecture, and inference procedure unchanged. It only modifies the training objective so that the graph-free student does not miss reliable teacher knowledge disproportionately for particular labels.

Standard GNN-to-MLP distillation encourages the student to match the teacher on average. However, average agreement does not reveal whether teacher knowledge is inherited uniformly across labels. `TED` reframes distillation as the transfer of reliable teacher knowledge rather than the imitation of all teacher outputs. It first identifies teacher decisions that are valid transfer targets using correctness and confidence, then measures how much of this reliable knowledge the student fails to inherit for each label. The resulting objective penalizes only labels whose missed reliable transfer exceeds the average by more than a tolerance budget, directing optimization toward class-specific transfer failures that are not considered by average distillation losses. Figure 2 provides an overview of the distillation mechanism.

### 4.1 Reliable Teacher Decisions

Let $f_\theta$ be the frozen GNN teacher. For each labeled training node $v \in \mathcal{V}_L$, the teacher produces a predictive distribution $\mathbf{p}_v^t \in \Delta^{C-1}$ and predicted label $\hat{y}_v^t = \arg\max_{k \in \{1,...,C\}} \mathbf{p}_{v,k}^t$. We do not treat every teacher prediction as equally useful for transfer. If the teacher is incorrect or uncertain, enforcing agreement may propagate noise rather than useful graph-derived knowledge. We therefore define reliable teacher knowledge using both correctness and confidence.

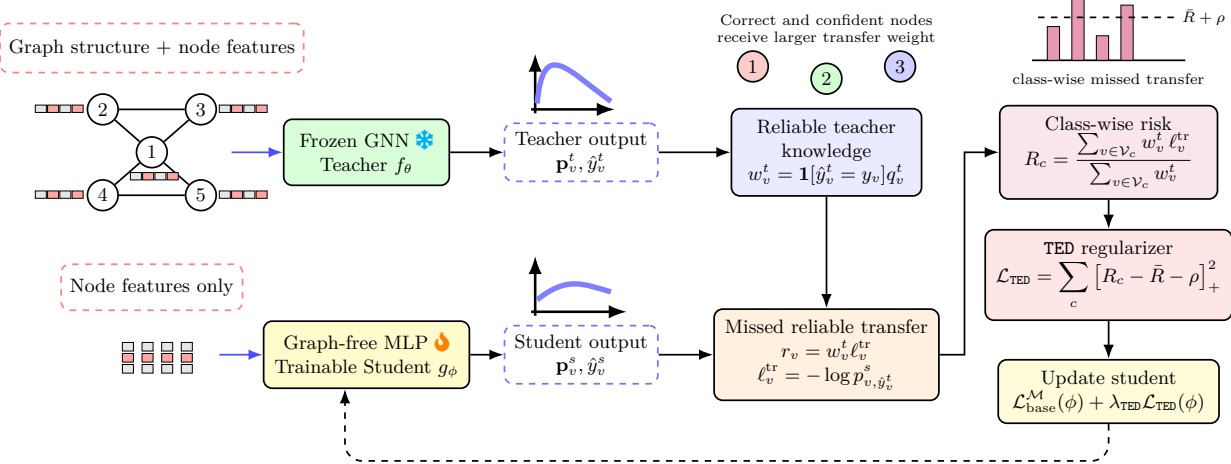

Figure 2: **Overview of `TED` framework**. (a) A frozen GNN teacher uses graph structure and node features, while the student MLP receives only node features. (b) `TED` identifies reliable teacher knowledge using correctness and confidence, and computes node-level missed reliable transfer. (c) These quantities are aggregated class-wise, and classes with excessive missed transfer are penalized through the `TED` regularizer.

Correctness is available on labeled training nodes through the indicator $\mathbf{1}[\hat{y}_v^t = y_v]$. Confidence is measured by normalized teacher certainty:

$$q_v^t = 1 - \frac{\mathcal{H}(\mathbf{p}_v^t)}{\log C}, \qquad \mathcal{H}(\mathbf{p}_v^t) = -\sum_{k=1}^{C} \mathbf{p}_{v,k}^t \log \mathbf{p}_{v,k}^t. \tag{6}$$

Since $0 \leq \mathcal{H}(\mathbf{p}_v^t) \leq \log C$, we have $q_v^t \in [0, 1]$. The score is large when the teacher distribution is concentrated and small when the teacher prediction is uncertain. The reliability weight of node $v$ is

$$w_v^t = \mathbf{1}[\hat{y}_v^t = y_v]\, q_v^t. \tag{7}$$

Thus, incorrect teacher predictions receive zero weight, while correct but uncertain predictions are downweighted. The transfer-equity objective therefore focuses on teacher decisions that are both valid and reliable.

### 4.2 Class-Conditional Missed Transfer Risk

Valid Transfer Agreement (Definition 1) measures whether the student preserves correct teacher decisions, but it is defined through hard predicted labels and is therefore not directly differentiable. To optimize the same transfer behavior during training, we introduce a differentiable teacher-targeted surrogate. Let $\mathbf{p}_v^s = g_\phi(\mathbf{x}_v)$ denote the student prediction. For a labeled training node $v$, we define

$$\ell_v^{\mathrm{tr}}(\phi) = -\log \mathbf{p}_{v,\hat{y}_v^t}^s. \tag{8}$$

This loss penalizes the student when it assigns low probability to the teacher's predicted class. Unlike entropy minimization, which can make the student confident in any class, $\ell_v^{\mathrm{tr}}$ anchors the desired confidence to the teacher decision being transferred. We combine this surrogate with the reliability weight $w_v^t$ to obtain the node-level missed reliable transfer risk

$$r_v(\phi) = w_v^t \ell_v^{\mathrm{tr}}(\phi). \tag{9}$$

This quantity is large only when the teacher decision is reliable and the student assigns insufficient probability to it. For each class $c$, we then define the class-conditional missed transfer risk as:

$$R_c(\phi) = \frac{\sum_{v \in \mathcal{V}_c} w_v^t \ell_v^{\mathrm{tr}}(\phi)}{\sum_{v \in \mathcal{V}_c} w_v^t}. \tag{10}$$

In implementation, we include a small constant in the denominator to prevent numerical instability. The risk $R_c(\phi)$ estimates how much reliable teacher knowledge for class $c$ is missed by the graph-free student. High values indicate that the teacher makes correct and confident decisions for that class, yet the student fails to place sufficient probability on those decisions. This makes $R_c(\phi)$ a differentiable training proxy to the class-level missed transfer captured by VTA.

### 4.3 Class-Conditional Missed-Transfer Control

The class-conditional risks $\{R_c(\phi)\}_{c \in \mathcal{C}_{\mathrm{rel}}}$ measure how much reliable teacher knowledge the graph-free student fails to inherit for each label during training. Standard GNN-to-MLP distillation objectives optimize aggregate student performance or average teacher–student agreement, but they do not control how missed transfer is distributed across labels. As a result, a student may preserve the teacher well on average while losing reliable teacher knowledge disproportionately for a subset of classes.

We formulate `TED` as an auxiliary objective for an arbitrary graph-free distillation method. Let $\mathcal{M}$ denote a base GNN-to-MLP distillation method, and let $\mathcal{L}_{\mathrm{base}}^{\mathcal{M}}(\phi)$ denote the original objective used to train its student. This objective may include supervised classification, soft-label distillation, reliability weighting, uncertainty modeling, fairness regularization, or other method-specific components. `TED` leaves this base objective unchanged and adds a class-conditional term that penalizes excessive missed transfer of reliable teacher knowledge.

We first define the set of labels with nonzero reliable teacher mass as $\mathcal{C}_{\mathrm{rel}} = \left\{ c : \sum_{v \in \mathcal{V}_c} w_v^t > 0 \right\}$. The average missed transfer risk over these labels is

$$\bar{R}(\phi) = \frac{1}{|\mathcal{C}_{\mathrm{rel}}|} \sum_{c \in \mathcal{C}_{\mathrm{rel}}} R_c(\phi). \tag{11}$$

For class $c$, the excess missed transfer risk is $R_c(\phi) - \bar{R}(\phi)$, which measures how much more reliable teacher knowledge is missed for that class relative to the average label.

The desired class-conditional transfer behavior is that no label should have excess missed transfer beyond a tolerance budget $\rho \geq 0$. This gives the bounded excess-risk condition

$$R_c(\phi) - \bar{R}(\phi) \leq \rho, \qquad \forall c \in \mathcal{C}_{\mathrm{rel}}. \tag{12}$$

Equivalently, $\rho$ controls how much class-level variation in missed reliable transfer is permitted. Smaller values impose stricter transfer uniformity, whereas larger values allow more heterogeneity across labels.

This condition yields the following ideal constrained formulation for a base method $\mathcal{M}$:

$$\begin{aligned} \min_{\phi} \quad & \mathcal{L}_{\mathrm{base}}^{\mathcal{M}}(\phi) \\ \text{s.t.} \quad & R_c(\phi) - \bar{R}(\phi) \leq \rho, \qquad \forall c \in \mathcal{C}_{\mathrm{rel}}. \end{aligned} \tag{13}$$

This formulation separates the role of the base distillation method from the class-conditional transfer requirement. The base objective preserves the original training criterion of $\mathcal{M}$, while the constraint rules out solutions in which reliable teacher knowledge is missed substantially more for some labels than for others.

Several standard constrained optimization tools could be used to address Eq. (13), including primal–dual methods, augmented Lagrangian methods, and penalty relaxations (Bertsekas, 1997; Boyd & Vandenberghe, 2004; Nocedal & Wright, 2006). We adopt a penalty relaxation because it introduces no dual variables, requires only a differentiable class-wise penalty term, and can be added to existing graph-free distillation objectives without changing their optimization pipeline. Comparison with alternative optimization approaches is beyond the scope of this work and is left for future work.

Specifically, TED uses a squared hinge penalty on violations of Eq. (12):

$$\mathcal{L}_{\mathtt{TED}}(\phi) = \frac{1}{|\mathcal{C}_{\mathrm{rel}}|} \sum_{c \in \mathcal{C}_{\mathrm{rel}}} \left[ R_c(\phi) - \bar{R}(\phi) - \rho \right]_+^2, \tag{14}$$

where $[a]_+ = \max\{a, 0\}$. The hinge form makes the penalty selective. Labels whose missed transfer risk lies within the tolerance budget do not contribute to the penalty, while labels that violate the budget contribute according to the squared magnitude of the violation.

The final objective for method $\mathcal{M}$ augmented with TED is:

$$\mathcal{L}_{\text{TED}+\mathcal{M}}(\phi) = \mathcal{L}_{\text{base}}^{\mathcal{M}}(\phi) + \lambda_{\text{TED}}\mathcal{L}_{\text{TED}}(\phi), \tag{15}$$

where $\lambda_{\text{TED}} \geq 0$ controls the strength of the class-conditional transfer penalty. Thus, TED is a differentiable relaxation of the bounded excess-risk formulation. Since all TED-specific quantities are used only during training, the deployed student remains identical to the graph-free student of the base method and requires no graph access at inference.

## 4.4 Optimization and Integration

Training with TED uses standard gradient-based optimization. Since the teacher is fixed, its predictions and reliability weights can be precomputed on the labeled nodes. The detailed training procedure is summarized in Algorithm 1.

**Instantiation with existing methods.** For GLNN (Zhang et al., 2021), $\mathcal{M} = $ GLNN and $\mathcal{L}_{\text{base}}^{\mathcal{M}}$ is the original supervised and soft-label distillation objective used to train the MLP student from the GNN teacher. Adding TED gives the augmented objective

$$\mathcal{L}_{\text{TED}+\text{GLNN}}(\phi) = \alpha\mathcal{L}_{\text{CE}}(\phi) + (1-\alpha)\sum_{v\in\mathcal{V}} D_{\text{KL}}\left(\mathbf{p}_{v,\tau}^t \,\|\, \mathbf{p}_{v,\tau}^s\right) + \lambda_{\text{TED}}\frac{1}{|\mathcal{C}_{\text{rel}}|}\sum_{c\in\mathcal{C}_{\text{rel}}}\left[R_c(\phi) - \bar{R}(\phi) - \rho\right]_+^2 \tag{16}$$

where $\mathcal{L}_{CE}(\phi)$ is the cross-entropy loss, $\tau$ is the distillation temperature, and $D_{\text{KL}}(\cdot)$ is the Kullback–Leibler (KL) divergence. Additional instantiations follow the same augmented-objective form and are detailed in the Appendix D.

## 4.5 Objective Analysis

We provide a connection between the differentiable objective optimized by TED and the hard class-wise transfer metrics used for evaluation. For each class $c \in \mathcal{C}_{\text{valid}}$, recall that $\mathcal{T}_c = \{v \in \mathcal{V}_c : \hat{y}_v^t = y_v\}$ is the set of teacher-correct nodes for class $c$. Define the hard missed-transfer rate associated with VTA as

$$U_c = 1 - \text{VTA}_c = \frac{1}{|\mathcal{T}_c|}\sum_{v\in\mathcal{T}_c}\mathbf{1}[\hat{y}_v^s \neq \hat{y}_v^t]. \tag{17}$$

Since $U_c = 1 - \text{VTA}_c$, the Equity Gap can equivalently be written as

$$\text{EG} = \max_{c\in\mathcal{C}_{\text{valid}}} U_c - \min_{c\in\mathcal{C}_{\text{valid}}} U_c. \tag{18}$$

Let us define $\gamma_c = \min_{v\in\mathcal{T}_c} q_v^t$ and $\bar{q}_c^t = \frac{1}{|\mathcal{T}_c|}\sum_{v\in\mathcal{T}_c} q_v^t$, where $\gamma_c$ is the minimum teacher confidence among teacher-correct nodes of class $c$, and $\bar{q}_c^t$ is the corresponding average confidence.

**Proposition 1.** *For any class $c \in \mathcal{C}_{\text{valid}}$ with $\gamma_c > 0$,*

$$1 - \text{VTA}_c = U_c \leq \frac{\bar{q}_c^t}{\gamma_c \log 2}R_c(\phi). \tag{19}$$

*Proof.* The proof is provided in Appendix B. $\square$

Proposition 1 establishes a connection between the differentiable missed-transfer risk $R_c(\phi)$ and the hard missed-transfer rate underlying VTA. This motivates $R_c(\phi)$ as a training-time surrogate for the class-wise transfer failures that VTA measures after optimization. All reported transfer disparity results are computed directly using the post-training VTA and Equity Gap metrics.

**Convergence and Time Complexity.** Under the standard smoothness conditions used in nonconvex optimization, the usual stationarity guarantee applies (Phuong & Lampert, 2019; Garrigos & Gower, 2023). The convergence behavior and time complexity of TED are detailed in Appendix C.

# 5 Experiments

We evaluate TED on real-world graph benchmarks to test whether it reduces class-conditional missed transfer while preserving the predictive and deployment benefits of graph-free GNN-to-MLP distillation. Specifically, we ask: **RQ1: Does TED reduce transfer disparity while preserving predictive utility across teachers and settings? RQ2: How does TED compare with reweighting and robust-optimization baselines? RQ3: Does TED remain effective on heterophilic graphs? RQ4: What training-time overhead does TED introduce, and does it preserve graph-free inference? RQ5: How important are the different components of TED? RQ6: What class-level factors explain disparities in graph-free distillation?**

**Experimental Settings and Baselines.**

**Datasets.** We evaluate TED on eight node-classification datasets spanning homophilic and heterophilic graph regimes. The homophilic datasets are Cora (McCallum et al., 2000), Citeseer (Giles et al., 1998), Pubmed (McCallum et al., 2000), Amazon-Photo, Coauthor-CS, and Coauthor-Physics (Shchur et al., 2018). The heterophilic suite includes Wisconsin and Cornell (Lim et al., 2021). Additional dataset statistics are provided in the Appendix E. Each experiment is repeated over five independent random seeds, and we report the mean and standard deviation. For Cora, Citeseer, and Pubmed, we use the standard citation-network splits introduced by (Kipf & Welling, 2016). For Coauthor-CS, Coauthor-Physics, and Amazon-Photo, we follow the protocol adopted in prior graph-free distillation studies (Zhang et al., 2021). For the heterophilic datasets Wisconsin and Cornell, we use the public splits provided by (Pei et al., 2020).

**Evaluation Settings.** We evaluate all methods under both transductive and inductive settings. Let the node set be partitioned as $\mathcal{V} = \mathcal{V}_L \sqcup \mathcal{V}_U^{\text{obs}} \sqcup \mathcal{V}_U^{\text{ind}}$, where $\mathcal{V}_L$ denotes labeled training nodes, $\mathcal{V}_U^{\text{obs}}$ denotes unlabeled nodes observed during training, and $\mathcal{V}_U^{\text{ind}}$ denotes nodes reserved for inductive evaluation. For any subset $S \subseteq \mathcal{V}$, we write $\mathbf{X}_S = \{\mathbf{x}_v : v \in S\}$ and $\mathbf{Y}_S = \{y_v : v \in S\}$.

In the transductive setting, the full graph structure and all node features are available during training, while supervision is provided only on $\mathcal{V}_L$. Models are evaluated on the unlabeled evaluation nodes $\mathcal{V}_U^{\text{obs}} \cup \mathcal{V}_U^{\text{ind}}$, depending on the dataset split. In the inductive setting, nodes in $\mathcal{V}_U^{\text{ind}}$ are held out during training. Training uses the observed subgraph induced by $\mathcal{V}_L \cup \mathcal{V}_U^{\text{obs}}$, together with labels only from $\mathcal{V}_L$. Evaluation is performed on $\mathcal{V}_U^{\text{ind}}$. Both evaluation protocols follow prior GNN-to-MLP distillation studies (Singh et al., 2025; Wu et al., 2023; Zhang et al., 2021).

**Baselines.** We compare TED-augmented methods with representative graph-free distillation baselines, including the GNN teacher, a standalone MLP, GLNN (Zhang et al., 2021), KRD (Wu et al., 2023), and FAITH (Singh et al., 2026). Following, Zhang et al. (2021), we use GraphSAGE (Hamilton et al., 2017) as the default teacher architecture. We further evaluate GCN (Kipf & Welling, 2016) to test robustness to the teacher architecture. For heterophilic datasets, we use ACM-GCN (Luan et al., 2022) as the teacher.

**Evaluation Metrics.** We evaluate predictive utility using average classification accuracy and transfer disparity using the Equity Gap in Definition 2. To complement the scalar Equity Gap metric, Appendix J reports complementary diagnostics: MeanVTA summarizes the average level of valid teacher-knowledge inheritance across classes, WorstVTA captures transfer to the least-inherited class, and StdVTA measures dispersion across class-wise VTA values. We also report the teacher-correct support for each class.

## 5.1 Results

**Effect on Node Classification (RQ1).** Table 1 reports transductive and inductive results on homophilic datasets using GraphSAGE as the teacher. The results show that strong average accuracy does not necessarily imply uniform transfer of teacher knowledge. Across GLNN, KRD, and FAITH, the base graph-free students achieve competitive accuracy, yet their Equity Gap reveals substantial class-conditional missed transfer. Adding TED consistently reduces Equity Gap across all three base distillation approaches while preserving competitive predictive utility. For each TED variant, we use the training-time tolerance budget $\rho = 10\%$. In the transductive setting, TED$_{+\text{GLNN}}$ reduces the macro-average Equity Gap from 13.98% to 8.79%, TED$_{+\text{KRD}}$ reduces it from 13.29% to 8.62%, and TED$_{+\text{FAITH}}$ reduces it from 17.67% to 11.55%. The corresponding macro-average accuracies remain comparable, changing from 85.86% to 85.70% for GLNN, from 86.00% to 86.04% for KRD, and from 85.43% to 86.30% for FAITH. The same trend holds in the inductive setting. TED$_{+\text{GLNN}}$ reduces the macro-average Equity Gap from 23.31% to 15.90%, TED$_{+\text{KRD}}$ reduces it from 21.22% to 16.36%, and TED$_{+\text{FAITH}}$ reduces it from 19.58% to 14.27%. Accuracy is again largely retained, changing

Table 1: Node classification accuracy and Equity Gap transductive and inductive settings, using GraphSAGE as the teacher. MLP and GraphSAGE are included as reference accuracy baselines. Parenthesized values in red report the relative Equity Gap reduction of each `TED`-augmented method with respect to its corresponding base method. Average rows report macro-averages over the datasets.

| Dataset | Reference Accuracy | | Metric | Graph-free Knowledge Distillation | | | | | |
|---|---|---|---|---|---|---|---|---|---|
| | MLP | SAGE | | GLNN | TED$_{+\text{GLNN}}$ | KRD | TED$_{+\text{KRD}}$ | FAITH | TED$_{+\text{FAITH}}$ |
| **Transductive Setting** | | | | | | | | | |
| Cora | 60.28±0.25 | 82.28±0.96 | Accuracy ↑ | 83.68±0.91 | 83.98±0.71 | 83.20±1.23 | 83.62±1.20 | 83.64±0.69 | 83.28±0.32 |
| | | | Equity Gap ↓ | 7.41±3.49 | 6.09±0.69 (17.81%) | 8.39±2.25 | 6.95±2.03 (17.16%) | 9.24±4.39 | 7.32±2.70 (20.78%) |
| Citeseer | 60.18±0.93 | 70.88±0.44 | Accuracy ↑ | 73.70±0.74 | 73.16±0.22 | 73.10±0.46 | 73.28±2.85 | 73.82±0.16 | 74.00±0.44 |
| | | | Equity Gap ↓ | 33.17±21.14 | 11.41±1.39 (65.60%) | 29.95±22.34 | 10.72±2.32 (64.21%) | 27.95±7.55 | 15.74±5.63 (43.69%) |
| Pubmed | 72.46±1.32 | 77.32±0.60 | Accuracy ↑ | 79.86±0.70 | 81.08±0.32 | 80.50±0.58 | 80.74±0.51 | 81.36±0.36 | 81.98±0.27 |
| | | | Equity Gap ↓ | 5.16±1.92 | 4.39±2.33 (14.92%) | 6.07±1.43 | 4.48±2.11 (26.19%) | 8.80±1.59 | 6.19±1.64 (29.66%) |
| Photo | 77.86±1.68 | 90.33±1.68 | Accuracy ↑ | 90.63±1.66 | 89.43±2.40 | 92.17±1.48 | 91.78±1.58 | 92.59±1.59 | 92.59±1.50 |
| | | | Equity Gap ↓ | 23.43±7.78 | 18.26±5.03 (22.07%) | 20.42±5.85 | 17.91±4.23 (12.29%) | 27.63±15.87 | 17.57±7.05 (36.41%) |
| CS | 89.62±1.10 | 89.27±0.50 | Accuracy ↑ | 93.94±0.30 | 93.66±0.26 | 93.63±0.32 | 93.45±0.29 | 92.64±0.39 | 92.96±0.25 |
| | | | Equity Gap ↓ | 9.63±2.14 | 8.51±1.53 (11.63%) | 10.15±2.00 | 8.82±1.41 (13.10%) | 18.32±3.45 | 13.42±2.59 (26.75%) |
| Physics | 87.45±2.65 | 91.09±0.88 | Accuracy ↑ | 93.33±0.89 | 92.86±0.93 | 93.41±0.92 | 93.36±1.27 | 88.51±9.08 | 92.98±0.90 |
| | | | Equity Gap ↓ | 5.05±1.73 | 4.09±1.46 (19.01%) | 4.78±2.09 | 2.82±1.05 (41.00%) | 14.06±13.68 | 9.05±3.93 (35.63%) |
| Macro Avg. | **74.64** | **83.53** | Accuracy ↑ | **85.86** | **85.70** | **86.00** | **86.04** | **85.43** | **86.30** |
| | | | Equity Gap ↓ | **13.98** | **8.79** (37.12%) | **13.29** | **8.62** (35.18%) | **17.67** | **11.55** (34.63%) |
| **Inductive Setting** | | | | | | | | | |
| Cora | 60.58±0.43 | 79.76±0.24 | Accuracy ↑ | 72.54±2.38 | 73.16±0.71 | 72.14±2.52 | 73.04±0.78 | 72.42±1.56 | 72.12±1.91 |
| | | | Equity Gap ↓ | 41.67±24.95 | 26.81±3.71 (35.66%) | 40.20±24.27 | 27.65±3.36 (31.22%) | 32.33±11.34 | 22.94±6.39 (29.04%) |
| Citeseer | 60.40±0.44 | 70.08±0.63 | Accuracy ↑ | 70.54±0.88 | 71.22±0.89 | 70.76±0.62 | 70.40±0.58 | 71.02±1.09 | 71.24±0.68 |
| | | | Equity Gap ↓ | 32.26±9.98 | 19.94±3.17 (38.19%) | 25.66±2.57 | 20.76±4.08 (19.10%) | 26.08±5.20 | 16.52±1.68 (36.66%) |
| Pubmed | 73.32±1.16 | 78.16±0.42 | Accuracy ↑ | 80.94±0.90 | 80.80±0.37 | 80.84±0.48 | 81.03±0.31 | 80.32±1.00 | 81.00±0.59 |
| | | | Equity Gap ↓ | 7.21±1.36 | 5.73±2.43 (20.53%) | 6.25±1.67 | 5.09±2.10 (18.56%) | 5.22±2.04 | 2.03±1.53 (61.11%) |
| Photo | 78.33±1.95 | 88.70±2.08 | Accuracy ↑ | 89.52±2.34 | 89.32±2.37 | 89.01±2.00 | 89.68±2.14 | 89.01±1.88 | 88.70±1.69 |
| | | | Equity Gap ↓ | 31.28±9.00 | 21.42±7.18 (31.52%) | 26.21±5.11 | 22.93±2.21 (12.51%) | 27.48±11.87 | 24.22±8.26 (11.86%) |
| CS | 89.93±1.06 | 89.16±0.37 | Accuracy ↑ | 92.57±0.41 | 92.43±0.24 | 92.51±0.39 | 92.77±0.42 | 92.32±0.57 | 92.27±0.49 |
| | | | Equity Gap ↓ | 20.25±2.62 | 17.07±5.22 (15.70%) | 22.47±1.79 | 17.77±1.11 (20.92%) | 19.76±4.51 | 15.77±2.22 (20.19%) |
| Physics | 87.60±1.75 | 90.89±2.15 | Accuracy ↑ | 93.04±1.48 | 92.75±1.64 | 93.10±1.44 | 92.82±1.80 | 91.74±2.56 | 92.52±1.42 |
| | | | Equity Gap ↓ | 7.19±3.05 | 4.40±0.85 (38.80%) | 6.55±2.88 | 3.95±0.90 (39.69%) | 6.59±0.62 | 4.14±0.98 (37.18%) |
| Macro Avg. | **75.03** | **82.79** | Accuracy ↑ | **83.19** | **83.28** | **83.06** | **83.29** | **82.81** | **82.98** |
| | | | Equity Gap ↓ | **23.31** | **15.90** (31.81%) | **21.22** | **16.36** (22.92%) | **19.58** | **14.27** (27.11%) |

Table 2: Comparison of `TED`$_{+\text{GLNN}}$ against reweighting and robust objectives using accuracy and Equity Gap. For each method, the red subscript reports the percentage change relative to GLNN. Up and down arrows indicate whether the metric value increases or decreases, while a horizontal arrow indicates no change. Standard deviations are provided in the Appendix.

| Dataset | GLNN | | Balanced KD | | DRO | | Focal Loss | | TED | |
|---|---|---|---|---|---|---|---|---|---|---|
| | Acc.↑ | EG↓ | Acc.↑ | EG↓ | Acc.↑ | EG↓ | Acc.↑ | EG↓ | Acc.↑ | EG↓ |
| Cora | 83.68 | 7.41 | 83.74$_{(0.07\%\uparrow)}$ | 8.39$_{(13.23\%\uparrow)}$ | 83.34$_{(0.41\%\downarrow)}$ | 8.12$_{(9.58\%\uparrow)}$ | 83.28$_{(0.48\%\downarrow)}$ | 8.92$_{(20.38\%\uparrow)}$ | 83.98$_{(0.36\%\uparrow)}$ | 6.09$_{(17.81\%\downarrow)}$ |
| CiteSeer | 73.70 | 33.17 | 72.66$_{(1.41\%\downarrow)}$ | 64.89$_{(95.63\%\uparrow)}$ | 72.32$_{(1.87\%\downarrow)}$ | 11.65$_{(64.88\%\downarrow)}$ | 73.54$_{(0.22\%\downarrow)}$ | 32.99$_{(0.54\%\downarrow)}$ | 73.16$_{(0.73\%\downarrow)}$ | 11.41$_{(65.60\%\downarrow)}$ |
| Pubmed | 79.86 | 5.16 | 80.86$_{(1.25\%\uparrow)}$ | 6.23$_{(20.74\%\uparrow)}$ | 80.40$_{(0.68\%\uparrow)}$ | 5.82$_{(12.79\%\uparrow)}$ | 81.00$_{(1.43\%\uparrow)}$ | 5.19$_{(0.58\%\uparrow)}$ | 81.08$_{(1.53\%\uparrow)}$ | 4.39$_{(14.92\%\downarrow)}$ |
| CS | 93.94 | 9.63 | 93.46$_{(0.51\%\downarrow)}$ | 9.60$_{(0.31\%\downarrow)}$ | 92.41$_{(1.63\%\downarrow)}$ | 9.01$_{(6.44\%\downarrow)}$ | 93.45$_{(0.52\%\downarrow)}$ | 10.22$_{(6.13\%\uparrow)}$ | 93.66$_{(0.30\%\downarrow)}$ | 8.51$_{(11.63\%\downarrow)}$ |
| Physics | 93.33 | 5.05 | 93.06$_{(0.29\%\downarrow)}$ | 7.07$_{(40.00\%\uparrow)}$ | 90.59$_{(2.94\%\downarrow)}$ | 5.85$_{(15.84\%\uparrow)}$ | 93.33$_{(0.00\%\leftrightarrow)}$ | 6.35$_{(25.74\%\uparrow)}$ | 92.86$_{(0.50\%\downarrow)}$ | 4.09$_{(19.01\%\downarrow)}$ |
| Amazon-Photo | 90.63 | 23.43 | 91.91$_{(1.41\%\uparrow)}$ | 25.72$_{(9.77\%\uparrow)}$ | 81.22$_{(10.38\%\downarrow)}$ | 19.47$_{(16.90\%\downarrow)}$ | 91.98$_{(1.49\%\uparrow)}$ | 21.60$_{(7.81\%\downarrow)}$ | 89.43$_{(1.32\%\downarrow)}$ | 18.26$_{(22.07\%\downarrow)}$ |
| Macro Avg. | **85.86** | **13.98** | **85.95**$_{(0.10\%\uparrow)}$ | **20.32**$_{(45.35\%\uparrow)}$ | **83.38**$_{(2.89\%\downarrow)}$ | **9.99**$_{(28.54\%\downarrow)}$ | **86.10**$_{(0.28\%\uparrow)}$ | **14.21**$_{(1.65\%\uparrow)}$ | **85.70**$_{(0.19\%\downarrow)}$ | **8.79**$_{(37.12\%\downarrow)}$ |

from 83.19% to 83.28% for GLNN, from 83.06% to 83.29% for KRD, and from 82.81% to 82.98% for FAITH. These results indicate that class-conditional missed transfer is not specific to a single distillation objective. It appears in standard graph-free KD methods as well as in FAITH, a fairness-oriented distillation baseline. `TED` provides a method-agnostic training-time correction that reduces disparity in the inheritance of correct teacher decisions while keeping the deployed model as the same graph-free MLP student. Results with additional teacher architecture is reported in Appendix G.

**Comparison with Reweighting and Robust-Optimization Baselines (RQ2).** Table 2 shows that reweighting and robust-optimization baselines do not consistently mitigate class-conditional missed transfer.

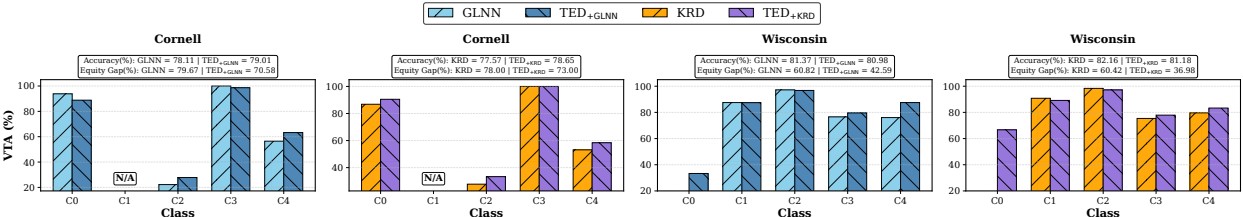

Figure 3: Class-wise Valid Transfer Agreement and Equity gap evaluation on heterophilic datasets. We compare the base GLNN and KRD distillation methods with their corresponding TED-augmented variants. TED-augmented students improves the VTA of the most under-transferred class under vanilla distillation.

Class-Balanced KD (Zhang et al., 2023) preserves the macro-average accuracy of GLNN, but increases the macro-average EG by 45.35%, with particularly large degradation on CiteSeer and Amazon-Photo. Focal Loss (Tan et al., 2022) achieves the highest macro-average accuracy, but increases EG slightly by 1.65%, suggesting that student-side hard-example emphasis is not sufficient to address transfer disparity. DRO (Vilouras et al., 2023) reduces macro-average EG by 28.54%, but this comes with a larger accuracy drop of 2.89%, especially on Amazon-Photo. In contrast, TED provides the most favorable tradeoff. It reduces EG on every dataset and achieves the largest macro-average EG reduction of 37.12% relative to GLNN, while loosing macro-average accuracy by only 0.19%. This supports that class-conditional missed transfer is better addressed by a transfer-specific residual based on reliable teacher decisions, as used in TED.

**Performance on Heterophilic Datasets (RQ3).** We further evaluate TED on heterophilic datasets. For these datasets, we use ACM-GCN as the teacher architecture. Figure 3 shows that transfer disparity also arises on heterophilic graphs. The base graph-free distillation methods achieve competitive accuracy, but their Equity Gap and class-wise VTA reveal that correct teacher decisions are not inherited uniformly across labels. Adding TED reduces this disparity for the corresponding base methods while maintaining comparable predictive utility. On the Wisconsin dataset, the TED-augmented variants show a clear improvement in worst-class VTA. The improvement is reflected not only in lower Equity Gap, but also in improved worst-class VTA across the reported settings. In the public Cornell split, class C1 contains only one sample overall and no test instance. Because VTA is computed on test samples for which the teacher is correct, C1 has no evaluable transfer support. We therefore mark it as N/A in Figure 3 and exclude it from the Equity Gap calculation. Consistent with Definition 2, we compute Equity Gap only over valid classes. Under this evaluation, TED improves the worst-class VTA on Cornell dataset as well.

**Computational Time (RQ4).** We report runtime on PubMed dataset to verify that the gains in transfer equity do not change the deployment profile of graph-free distillation. Figure 4 shows that TED introduces only a small training-time overhead due to its class-wise aggregation and regularization term. At inference time, the runtime matches the corresponding graph-free base method. Labels L1, L2 and L3 indicate the number of layers in SAGE.

**Ablation Study (RQ5).** We conduct an ablation study to assess the contribution of the main components of TED. Starting from $TED_{+GLNN}$, we consider two simplified variants: one without reliability weighting (w/o RW), which removes the teacher-confidence weighting, and one without teacher correctness filtering (w/o TC), which no longer excludes incorrect teacher predictions from the missed-transfer estimate. The results in Figure 5 show the positive contribution of each component under inductive setting. Similar trends are observed in the transductive setting, as reported in the Appendix H.

**Diagnosing Class-Wise Transfer Disparities (RQ6).** To better understand why some labels suffer larger transfer failures, we examine whether low-VTA classes also have weaker conditions for graph-free recovery. We consider three class-level diagnostics. First, we report the test-set class support share, $\frac{|\mathcal{V}_c|}{|\mathcal{V}|}$, which captures how much data is available for class $c$. Second, we compute class-wise edge homophily (Platonov et al., 2023), $h_c = \frac{|\{(u,v)\in\mathcal{E}:y_u=y_v=c\}|}{|\{(u,v)\in\mathcal{E}:y_v=c \text{ or } y_u=c\}|}$, which measures the fraction of same-label edges as-

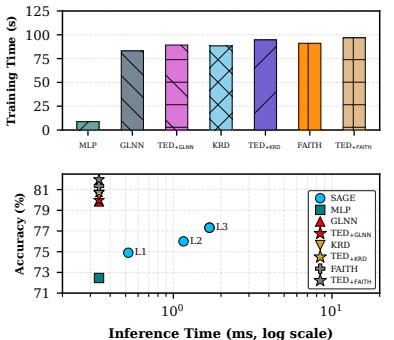

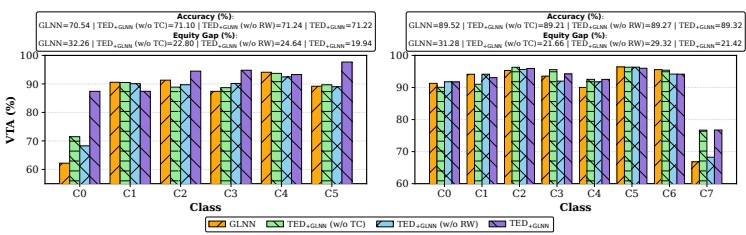

Figure 5: Ablation analysis of TED on CiteSeer (left) and Amazon-Photo (right) show that removing reliability weighting or teacher-correctness filtering weakens the reduction in class-conditional transfer disparity. In both datasets, TED also improves the VTA of the most under-transferred class under vanilla GLNN

Figure 4: Training time (top). Inference time and accuracy comparison (bottom).

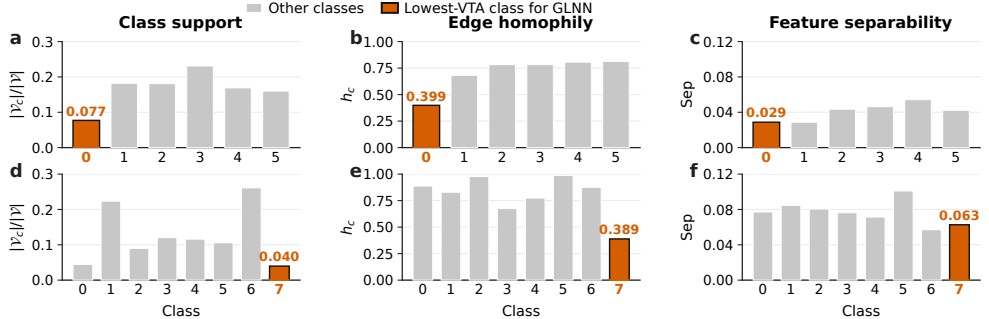

Figure 6: Class support, edge homophily, and feature separability comparison for CiteSeer (top) and Amazon-Photo (bottom). The lowest-VTA class under vanilla GLNN is highlighted in orange.

sociated with class $c$. Third, we quantify raw-feature separability after normalization. Let $\tilde{x}_v = x_v/\|x_v\|_2$ denote the normalized feature vector of node $v$, and let $\mu_c = \frac{1}{|\mathcal{V}_c|}\sum_{v\in\mathcal{V}_c}\tilde{x}_v$ be the centroid of class $c$. We define the within-class spread as $\mathrm{Intra}(c) = \frac{1}{|\mathcal{V}_c|}\sum_{v\in\mathcal{V}_c}\|\tilde{x}_v - \mu_c\|_2^2$, and the nearest-centroid separation as $\mathrm{Inter}_{\min}(c) = \min_{c'\neq c}\|\mu_c - \mu_{c'}\|_2^2$. We then report the bounded separability score $\mathrm{Sep}(c) = \frac{\mathrm{Inter}_{\min}(c)}{\mathrm{Inter}_{\min}(c)+\mathrm{Intra}(c)}$. Figure 6 highlights that, on both CiteSeer and Amazon-Photo, the highlighted class has low support, low edge homophily, and weak feature separability. These diagnostics suggest that uneven transfer is not explained by class frequency alone, but is associated with the combined effect of limited support, weak graph homophily, and poor feature-space separability. The results indicate that class-balanced reweighting alone may not be sufficient to fully address the Equity Gap. TED can address this gap more effectively because it directly regularizes the class-wise missed-transfer residual.

# 6 Conclusion

We investigated class-conditional transfer disparity in graph-free GNN-to-MLP distillation. Although distilled MLPs are commonly evaluated by average accuracy, we showed that they can fail to inherit valid GNN teacher knowledge uniformly across labels. We propose TED, a training-time objective augmentation that penalizes labels with excessive missed reliable transfer. TED preserves the student architecture and graph-free inference procedure, making it compatible with existing GNN-to-MLP distillation methods. Across graph benchmarks, TED reduces Equity Gap and improves worst-label transfer while maintaining competitive accuracy and efficient inference. These findings highlight the importance of evaluating graph-free distillation not only by aggregate utility, but also by the uniformity with which valid teacher knowledge is transferred across labels. A limitation of our approach is that reliability is estimated from labeled nodes, which may be restrictive in scarcely labeled settings.

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

# Appendix

## A TED Algorithm

Algorithm 1 presents the complete training procedure of the proposed TED framework.

---

**Algorithm 1** TED Training

---

**Require:** Frozen teacher $f_\theta$, student $g_\phi$, base method $\mathcal{M}$, tolerance $\rho$, regularization weight $\lambda_{\text{TED}}$
1: Precompute teacher labels $\hat{y}_v^t$ and reliability weights for labeled nodes $v \in \mathcal{V}_L$

$$w_v^t = \mathbf{1}[\hat{y}_v^t = y_v]\left(1 - \frac{\mathcal{H}(\mathbf{p}_v^t)}{\log C}\right)$$

2: **for** each training step **do**
3:   Compute the base objective $\mathcal{L}_{\text{base}}^{\mathcal{M}}(\phi)$
4:   Estimate $R_c(\phi)$ for each $c \in \mathcal{C}_{\text{rel}}$ using Eq. (10)
5:   Compute $\bar{R}(\phi) = |\mathcal{C}_{\text{rel}}|^{-1}\sum_{c \in \mathcal{C}_{\text{rel}}} R_c(\phi)$
6:   Compute $\mathcal{L}_{\text{TED}}(\phi)$ using Eq. (14)
7:   Update $\phi$ by gradient descent on $\mathcal{L}_{\text{TED}+\mathcal{M}}(\phi)$
8: **end for**
9: **return** $g_\phi$

---

## B Proof of Proposition 1

*Proof.* Fix a class $c \in \mathcal{C}_{\text{valid}}$ with $\gamma_c > 0$. Recall that $\mathcal{T}_c = \{v \in \mathcal{V}_c : \hat{y}_v^t = y_v\}$ is the set of teacher-correct nodes for class $c$. For every $v \in \mathcal{T}_c$, the teacher is correct, and therefore the reliability weight satisfies $w_v^t = q_v^t$. The teacher-targeted transfer loss is

$$\ell_v^{\text{tr}}(\phi) = -\log \mathbf{p}_{v,\hat{y}_v^t}^s. \tag{20}$$

We first establish a pointwise relation between hard teacher–student disagreement and the teacher-targeted loss. If $\hat{y}_v^s = \hat{y}_v^t$, then $\mathbf{1}[\hat{y}_v^s \neq \hat{y}_v^t] = 0$, and the desired pointwise bound holds trivially. If $\hat{y}_v^s \neq \hat{y}_v^t$, then the class predicted by the student has probability at least as large as the probability assigned to the teacher class. That is, for some $k \neq \hat{y}_v^t$,

$$\mathbf{p}_{v,k}^s \geq \mathbf{p}_{v,\hat{y}_v^t}^s. \tag{21}$$

Since the probabilities sum to one, this implies

$$\mathbf{p}_{v,\hat{y}_v^t}^s \leq \frac{1}{2}. \tag{22}$$

Hence,

$$-\log \mathbf{p}_{v,\hat{y}_v^t}^s \geq \log 2. \tag{23}$$

Therefore, for every $v \in \mathcal{T}_c$,

$$\mathbf{1}[\hat{y}_v^s \neq \hat{y}_v^t] \leq \frac{\ell_v^{\text{tr}}(\phi)}{\log 2}. \tag{24}$$

By definition, $\gamma_c = \min_{v \in \mathcal{T}_c} q_v^t$. Since $\gamma_c > 0$, we have $q_v^t \geq \gamma_c$ for all $v \in \mathcal{T}_c$. Because $\ell_v^{\text{tr}}(\phi) \geq 0$, it follows that

$$\ell_v^{\text{tr}}(\phi) \leq \frac{q_v^t}{\gamma_c}\ell_v^{\text{tr}}(\phi). \tag{25}$$

Combining Eq. (24) and Eq. (25), we obtain

$$\mathbf{1}[\hat{y}_v^s \neq \hat{y}_v^t] \leq \frac{q_v^t \ell_v^{\text{tr}}(\phi)}{\gamma_c \log 2}. \tag{26}$$

Averaging over $v \in \mathcal{T}_c$ gives

$$U_c = \frac{1}{|\mathcal{T}_c|} \sum_{v \in \mathcal{T}_c} \mathbf{1}[\hat{y}_v^s \neq \hat{y}_v^t] \leq \frac{1}{\gamma_c |\mathcal{T}_c| \log 2} \sum_{v \in \mathcal{T}_c} q_v^t \ell_v^{\mathrm{tr}}(\phi). \tag{27}$$

Since nodes outside $\mathcal{T}_c$ have zero reliability weight, the class-wise TED risk can be written as

$$R_c(\phi) = \frac{\sum_{v \in \mathcal{T}_c} q_v^t \ell_v^{\mathrm{tr}}(\phi)}{\sum_{v \in \mathcal{T}_c} q_v^t}. \tag{28}$$

Therefore,

$$\sum_{v \in \mathcal{T}_c} q_v^t \ell_v^{\mathrm{tr}}(\phi) = R_c(\phi) \sum_{v \in \mathcal{T}_c} q_v^t. \tag{29}$$

Substituting this identity yields

$$U_c \leq \frac{R_c(\phi) \sum_{v \in \mathcal{T}_c} q_v^t}{\gamma_c |\mathcal{T}_c| \log 2} = \frac{\bar{q}_c^t}{\gamma_c \log 2} R_c(\phi), \tag{30}$$

where

$$\bar{q}_c^t = \frac{1}{|\mathcal{T}_c|} \sum_{v \in \mathcal{T}_c} q_v^t. \tag{31}$$

Since $U_c = 1 - \mathrm{VTA}_c$, this proves

$$1 - \mathrm{VTA}_c = U_c \leq \frac{\bar{q}_c^t}{\gamma_c \log 2} R_c(\phi). \tag{32}$$

$\square$

## C  Convergence and Time Complexity

The TED term preserves the standard optimization structure of the base distillation objective. Most graph-free distillation losses such as cross-entropy and KL divergence are differentiable functions of the student logits. The TED penalty uses the squared hinge $[z]_+^2$, which is differentiable, composed with the class-wise risks $R_c(\phi)$. Thus, $\mathcal{L}_{\mathrm{TED}}(\phi)$ remains differentiable under the same regularity conditions as the teacher-targeted transfer loss. Under the standard smoothness and lower-boundedness assumptions used in nonconvex optimization, gradient descent on the augmented objective satisfies the usual stationarity guarantee. For completeness, we state the standard stationarity result for gradient descent on a smooth lower-bounded objective.

**Proposition 2.** *Suppose $\mathcal{L}_{\mathit{TED}_{+\mathcal{M}}}(\phi)$ is lower bounded and has $L$-Lipschitz continuous gradients along the optimization trajectory. If gradient descent is run with step size $0 < \eta \leq 1/L$, then after $T$ iterations,*

$$\min_{0 \leq t < T} \|\nabla \mathcal{L}_{\mathit{TED}_{+\mathcal{M}}}(\phi_t)\|_2^2 \leq \frac{2 \left( \mathcal{L}_{\mathit{TED}_{+\mathcal{M}}}(\phi_0) - \mathcal{L}_\star \right)}{\eta T}, \tag{33}$$

*where $\mathcal{L}_\star$ is a lower bound on $\mathcal{L}_{\mathit{TED}_{+\mathcal{M}}}(\phi)$.*

*Proof.* Since $\mathcal{L}_{\mathrm{TED}_{+\mathcal{M}}}(\phi)$ has $L$-Lipschitz continuous gradients along the optimization trajectory, the standard descent lemma implies that, for any two iterates $\phi_t$ and $\phi_{t+1}$,

$$\mathcal{L}_{\mathrm{TED}_{+\mathcal{M}}}(\phi_{t+1}) \leq \mathcal{L}_{\mathrm{TED}_{+\mathcal{M}}}(\phi_t) + \left\langle \nabla \mathcal{L}_{\mathrm{TED}_{+\mathcal{M}}}(\phi_t), \phi_{t+1} - \phi_t \right\rangle + \frac{L}{2} \|\phi_{t+1} - \phi_t\|_2^2. \tag{34}$$

Gradient descent updates the parameters as $\phi_{t+1} = \phi_t - \eta \nabla \mathcal{L}_{\mathrm{TED}_{+\mathcal{M}}}(\phi_t)$. Substituting this update into the descent inequality gives

$$\begin{aligned}
\mathcal{L}_{\mathrm{TED}_{+\mathcal{M}}}(\phi_{t+1}) &\leq \mathcal{L}_{\mathrm{TED}_{+\mathcal{M}}}(\phi_t) - \eta \left\| \nabla \mathcal{L}_{\mathrm{TED}_{+\mathcal{M}}}(\phi_t) \right\|_2^2 + \frac{L\eta^2}{2} \left\| \nabla \mathcal{L}_{\mathrm{TED}_{+\mathcal{M}}}(\phi_t) \right\|_2^2 \\
&= \mathcal{L}_{\mathrm{TED}_{+\mathcal{M}}}(\phi_t) - \eta \left( 1 - \frac{L\eta}{2} \right) \left\| \nabla \mathcal{L}_{\mathrm{TED}_{+\mathcal{M}}}(\phi_t) \right\|_2^2.
\end{aligned} \tag{35}$$

Since $0 < \eta \le 1/L$, we have $1 - \frac{L\eta}{2} \ge \frac{1}{2}$. Therefore,

$$\mathcal{L}_{\text{TED}+\mathcal{M}}(\phi_{t+1}) \le \mathcal{L}_{\text{TED}+\mathcal{M}}(\phi_t) - \frac{\eta}{2} \left\| \nabla \mathcal{L}_{\text{TED}+\mathcal{M}}(\phi_t) \right\|_2^2. \tag{36}$$

Rearranging yields

$$\left\| \nabla \mathcal{L}_{\text{TED}+\mathcal{M}}(\phi_t) \right\|_2^2 \le \frac{2}{\eta} \left( \mathcal{L}_{\text{TED}+\mathcal{M}}(\phi_t) - \mathcal{L}_{\text{TED}+\mathcal{M}}(\phi_{t+1}) \right). \tag{37}$$

Summing over $t = 0, \ldots, T-1$ gives

$$\sum_{t=0}^{T-1} \left\| \nabla \mathcal{L}_{\text{TED}+\mathcal{M}}(\phi_t) \right\|_2^2 \le \frac{2}{\eta} \sum_{t=0}^{T-1} \left( \mathcal{L}_{\text{TED}+\mathcal{M}}(\phi_t) - \mathcal{L}_{\text{TED}+\mathcal{M}}(\phi_{t+1}) \right)$$
$$= \frac{2}{\eta} \left( \mathcal{L}_{\text{TED}+\mathcal{M}}(\phi_0) - \mathcal{L}_{\text{TED}+\mathcal{M}}(\phi_T) \right). \tag{38}$$

Because $\mathcal{L}_\star$ is a lower bound on $\mathcal{L}_{\text{TED}+\mathcal{M}}(\phi)$, we have $\mathcal{L}_{\text{TED}+\mathcal{M}}(\phi_T) \ge \mathcal{L}_\star$. Hence,

$$\sum_{t=0}^{T-1} \left\| \nabla \mathcal{L}_{\text{TED}+\mathcal{M}}(\phi_t) \right\|_2^2 \le \frac{2}{\eta} \left( \mathcal{L}_{\text{TED}+\mathcal{M}}(\phi_0) - \mathcal{L}_\star \right). \tag{39}$$

Dividing by $T$ gives

$$\frac{1}{T} \sum_{t=0}^{T-1} \left\| \nabla \mathcal{L}_{\text{TED}+\mathcal{M}}(\phi_t) \right\|_2^2 \le \frac{2 \left( \mathcal{L}_{\text{TED}+\mathcal{M}}(\phi_0) - \mathcal{L}_\star \right)}{\eta T}. \tag{40}$$

Finally, the minimum is bounded by the average:

$$\min_{0 \le t < T} \left\| \nabla \mathcal{L}_{\text{TED}+\mathcal{M}}(\phi_t) \right\|_2^2 \le \frac{1}{T} \sum_{t=0}^{T-1} \left\| \nabla \mathcal{L}_{\text{TED}+\mathcal{M}}(\phi_t) \right\|_2^2. \tag{41}$$

Combining the last two inequalities proves

$$\min_{0 \le t < T} \left\| \nabla \mathcal{L}_{\text{TED}+\mathcal{M}}(\phi_t) \right\|_2^2 \le \frac{2 \left( \mathcal{L}_{\text{TED}+\mathcal{M}}(\phi_0) - \mathcal{L}_\star \right)}{\eta T}. \tag{42}$$

$\square$

TED introduces only a training-time overhead on top of the base distillation method. Since the teacher is frozen, teacher predictions and reliability weights can be precomputed once before student training. During each training epoch, TED computes the teacher-targeted losses and aggregates them by class to obtain the label-wise risks, which costs $\mathcal{O}(|\mathcal{V}_L| + C)$ for $|\mathcal{V}_L|$ labeled nodes and $C$ classes. Since node-classification benchmarks typically satisfy $|\mathcal{V}_L| \gg C$, the additional overhead is effectively dominated by $\mathcal{O}(|\mathcal{V}_L|)$ and is small relative to the forward and backward passes of the student.

## D  Integration with Existing GNN-to-MLP Methods

**Integration with KRD.**  Following KRD (Wu et al., 2023), let $\mathbf{h}_i^{(L)}$ denote the teacher GNN logits for node $i$ and let $\mathbf{z}_i^{(L)}$ denote the student MLP logits. KRD samples reliable knowledge points $j \in \mathcal{N}_i$ according to the reliability-based sampling distribution $p(s_j \mid \rho_i, \alpha^{(t)})$ and defines the reliable distillation loss as

$$\mathcal{L}_{\text{KRD}} = \mathbb{E}_i \mathop{\mathbb{E}}_{\substack{j \in \mathcal{N}_i \\ j \sim p(s_j | \rho_i, \alpha^{(t)})}} D_{\text{KL}} \left( \sigma(\mathbf{z}_j^{(L)}/\tau), \sigma(\mathbf{h}_i^{(L)}/\tau) \right), \tag{43}$$

where $\tau$ is the distillation temperature. The original KRD training objective is

$$\mathcal{L}_{\text{total}}^{\text{KRD}} = \frac{\lambda}{|\mathcal{V}_L|} \sum_{i \in \mathcal{V}_L} \mathcal{H} \left( y_i, \sigma(\mathbf{z}_i^{(L)}) \right) + (1 - \lambda) \left( \mathcal{L}_{\text{KD}} + \mathcal{L}_{\text{KRD}} \right). \tag{44}$$

We instantiate $\text{TED}_{+\text{KRD}}$ by adding the class-level transfer-equity regularizer to the original KRD objective:

$$\mathcal{L}_{\text{TED}+\text{KRD}} = \frac{\lambda}{|\mathcal{V}_L|} \sum_{i \in \mathcal{V}_L} \mathcal{H}\left(y_i, \sigma(\mathbf{z}_i^{(L)})\right) + (1 - \lambda)\left(\mathcal{L}_{\text{KD}} + \mathcal{L}_{\text{KRD}}\right) + \lambda_{\text{TED}} \frac{1}{|\mathcal{C}_{\text{rel}}|} \sum_{c \in \mathcal{C}_{\text{rel}}} \left[R_c(\phi) - \bar{R}(\phi) - \rho\right]_+^2. \tag{45}$$

**Integration with FAITH.** We instantiate $\text{TED}$ on top of FAITH (Singh et al., 2026) by adding the transfer-equity regularizer to the original FAITH training objective. We use FAITH as the base GNN-to-MLP distillation framework. Let $\mathbf{z}_v^s$ denote the student MLP logits for node $v$ and $\mathbf{z}_v^t$ denote the teacher GNN logits. FAITH optimizes a supervised classification loss, a soft-label distillation loss, a neighborhood-guided Dirichlet energy alignment loss, and an oracle similarity alignment loss:

$$\mathcal{L}_{\text{FAITH}}(\phi) = \alpha \sum_{v \in \mathcal{V}_L} \mathcal{L}_{\text{CE}}\left(\sigma(\mathbf{z}_v^s), y_v\right) + (1 - \alpha) \sum_{v \in \mathcal{V}} D_{\text{KL}}\left(\sigma(\mathbf{z}_v^s/\tau) \,\|\, \sigma(\mathbf{z}_v^t/\tau)\right) + \beta\mathcal{L}_{\text{DE}} + \gamma\mathcal{L}_{\text{IF}}. \tag{46}$$

Here, $\mathcal{L}_{\text{DE}}$ denotes the neighborhood-guided energy alignment loss, $\mathcal{L}_{\text{IF}}$ denotes the oracle similarity alignment loss, $\tau$ is the distillation temperature, and $\alpha, \beta, \gamma$ are the FAITH loss weights.

To obtain $\text{TED}+_{\text{FAITH}}$, we keep the FAITH objective unchanged and add the class-wise transfer-equity regularizer:

$$\mathcal{L}_{\text{TED}+\text{FAITH}}(\phi) = \mathcal{L}_{\text{FAITH}}(\phi) + \lambda_{\text{TED}}\mathcal{L}_{\text{TED}}(\phi) \tag{47}$$

where

$$\mathcal{L}_{\text{TED}}(\phi) = \frac{1}{|\mathcal{C}_{\text{rel}}|} \sum_{c \in \mathcal{C}_{\text{rel}}} \left[R_c(\phi) - \bar{R}(\phi) - \rho\right]_+^2. \tag{48}$$

Thus, $\text{TED}+_{\text{FAITH}}$ preserves the structural and fairness-aware objectives of FAITH while adding a constraint on class-wise missed transfer of reliable teacher knowledge. The added term is used only during training. At inference time, the deployed model remains the same graph-free MLP student used by FAITH.

## E Dataset Statistics

We utilize eight open graph datasets to validate the proposed $\text{TED}$ framework. An overview summary of statistics of the datasets is given in Table 3.

Table 3: Dataset statistics used to evaluate $\text{TED}$.

| Dataset | Type | # Nodes | # Edges | # Features | # Classes |
|---------|------|---------|---------|------------|-----------|
| Cora | Homophily | 2,708 | 5,278 | 1,433 | 7 |
| Citeseer | Homophily | 3,327 | 4,614 | 3,703 | 6 |
| Pubmed | Homophily | 19,717 | 44,324 | 500 | 3 |
| Photo | Homophily | 7,650 | 119,081 | 745 | 8 |
| CS | Homophily | 18,333 | 81,894 | 6,805 | 15 |
| Physics | Homophily | 34,493 | 247,962 | 8,415 | 5 |
| Wisconsin | Heterophily | 251 | 499 | 1,703 | 5 |
| Cornell | Heterophily | 183 | 295 | 1,703 | 5 |

## F Hyperparameter Details and System Configuration

For each baseline, we use the official implementation provided by the authors. For each $\text{TED}$-augmented variant, we retain the original implementation of the base method and tune its hyperparameters together with the $\text{TED}$-specific parameter using the validation split. The only additional hyperparameter introduced by $\text{TED}$ is the regularization strength $\lambda_{\text{TED}}$, which controls the contribution of the transfer-equity regularizer.

Table 4: Hyperparameter search space used in our experiments.

| Hyperparameter | Search space |
|---|---|
| Number of layers | $\{2, 3\}$ |
| Hidden dimension | $\{128, 256, 512, 1024, 2048\}$ |
| Learning rate | $\{0.001, 0.005, 0.01\}$ |
| Weight decay | $\{0, 0.0005, 0.001\}$ |

We select $\lambda_{\text{TED}}$ on the validation set from the range $[0, 1.5]$. We summarize the architectural and optimization hyperparameter search spaces in Table 4. All baseline methods and TED-augmented variants are implemented in PyTorch 1.6.0 (Paszke et al., 2019) using the DGL library (Wang et al., 2019). For each dataset and method, we select the hyperparameter configuration and training checkpoint that achieve the highest accuracy on the validation dataset. Experiments are run on a Linux machine equipped with an Intel Platinum 8360Y CPU and an NVIDIA A6000 GPU.

## G  Performance on GCN Teacher

To show the generalizability of TED with different teacher networks, we report accuracy and Equity Gap using GCN as teacher in both GLNN, KRD and FAITH. Results in Table 5 highlight that student MLPs augmented with TED consistently provide better tradeoff in utility and equity.

Table 5: Node classification accuracy and Equity Gap transductive and inductive settings, using GCN as the teacher. MLP and GCN are included as reference accuracy baselines. Parenthesized values in red report the relative Equity Gap reduction of each TED-augmented method with respect to its corresponding base method. Average rows report macro-averages over the datasets.

| Dataset | Reference Accuracy | | Metric | Graph-free Knowledge Distillation | | | | | |
|---|---|---|---|---|---|---|---|---|---|
| | **MLP** | **GCN** | | **GLNN** | TED$_{+\text{GLNN}}$ | **KRD** | TED$_{+\text{KRD}}$ | **FAITH** | TED$_{+\text{FAITH}}$ |
| | | | | **Transductive Setting** | | | | | |
| Cora | 60.28±0.25 | 81.48±1.64 | Accuracy ↑ | 83.10±1.11 | 83.30±2.02 | 83.34±0.65 | 83.24±1.57 | 83.34±0.94 | 83.66±1.03 |
| | | | Equity Gap ↓ | 9.87±2.01 | 7.78±3.44 (21.18%) | 10.08±2.22 | 7.59±2.06 (24.70%) | 7.17±2.55 | 5.03±1.12 (29.85%) |
| Citeseer | 60.18±0.93 | 71.60±0.23 | Accuracy ↑ | 73.32±0.86 | 73.62±0.42 | 73.12±0.69 | 73.54±0.57 | 74.68±0.52 | 74.88±0.37 |
| | | | Equity Gap ↓ | 33.38±28.26 | 14.38±5.70 (56.92%) | 41.26±27.05 | 13.49±6.90 (67.30%) | 17.84±6.55 | 13.26±4.76 (25.67%) |
| Pubmed | 72.46±1.32 | 77.16±0.45 | Accuracy ↑ | 81.38±0.71 | 81.06±0.46 | 80.82±1.02 | 81.16±0.45 | 82.14±0.55 | 81.88±0.64 |
| | | | Equity Gap ↓ | 6.40±2.32 | 5.04±3.13 (21.25%) | 5.78±2.24 | 4.79±2.27 (17.13%) | 3.81±1.90 | 2.64±1.36 (30.71%) |
| CS | 89.62±1.10 | 89.85±0.62 | Accuracy ↑ | 92.78±0.27 | 93.03±0.53 | 93.60±0.29 | 93.11±1.27 | 93.88±0.36 | 93.83±0.36 |
| | | | Equity Gap ↓ | 9.87±1.99 | 8.62±2.51 (12.66%) | 9.92±2.08 | 9.83±3.01 (0.91%) | 5.24±0.65 | 4.38±0.81 (16.41%) |
| Physics | 87.45±2.65 | 92.41±0.40 | Accuracy ↑ | 93.68±1.23 | 94.03±0.38 | 93.86±1.29 | 94.00±0.37 | 94.59±0.37 | 94.65±0.21 |
| | | | Equity Gap ↓ | 3.81±2.41 | 3.54±0.08 (7.09%) | 3.84±2.38 | 3.48±0.29 (9.38%) | 3.57±1.26 | 1.82±1.42 (49.02%) |
| Photo | 77.86±1.68 | 91.02±1.46 | Accuracy ↑ | 92.04±1.35 | 90.01±2.04 | 92.18±1.51 | 91.93±1.47 | 93.46±1.53 | 93.42±1.42 |
| | | | Equity Gap ↓ | 20.43±2.31 | 16.75±2.43 (18.01%) | 18.82±3.35 | 17.12±3.73 (9.03%) | 4.69±2.36 | 3.24±0.32 (30.92%) |
| Macro Avg. | **74.64** | **83.92** | Accuracy ↑ | **86.05** | **85.84** | **86.15** | **86.16** | **87.02** | **87.05** |
| | | | Equity Gap ↓ | **13.96** | **9.35** (33.02%) | **14.95** | **9.38** (37.26%) | **7.05** | **5.06** (28.23%) |
| | | | | **Inductive Setting** | | | | | |
| Cora | 60.58±0.43 | 79.24±0.59 | Accuracy ↑ | 73.64±0.48 | 73.00±1.01 | 73.64±0.14 | 73.10±0.30 | 73.24±0.35 | 73.06±0.26 |
| | | | Equity Gap ↓ | 30.96±2.58 | 24.59±2.16 (20.57%) | 29.40±2.83 | 26.04±1.33 (11.43%) | 28.24±9.26 | 26.75±3.09 (5.28%) |
| Citeseer | 60.40±0.44 | 72.20±0.24 | Accuracy ↑ | 71.62±0.74 | 71.70±0.57 | 71.50±0.52 | 71.26±0.16 | 70.72±0.57 | 70.34±0.76 |
| | | | Equity Gap ↓ | 53.31±21.07 | 23.16±3.96 (56.56%) | 53.36±21.35 | 22.15±3.88 (58.49%) | 61.00±22.07 | 19.89±5.04 (67.39%) |
| Pubmed | 73.32±1.16 | 77.58±0.28 | Accuracy ↑ | 81.02±0.39 | 80.98±0.31 | 81.06±0.62 | 80.94±0.28 | 80.20±0.20 | 80.32±0.72 |
| | | | Equity Gap ↓ | 5.66±1.80 | 2.80±1.40 (50.53%) | 4.46±2.54 | 3.95±2.44 (11.43%) | 5.29±2.69 | 3.30±3.35 (37.62%) |
| CS | 89.93±1.06 | 90.31±0.21 | Accuracy ↑ | 92.01±0.36 | 92.38±0.33 | 92.91±0.46 | 92.71±0.48 | 92.75±0.21 | 92.86±0.14 |
| | | | Equity Gap ↓ | 19.92±3.99 | 17.64±6.60 (11.45%) | 22.60±3.44 | 18.65±6.62 (17.48%) | 16.87±3.77 | 14.38±2.65 (14.76%) |
| Physics | 87.60±1.75 | 92.68±0.74 | Accuracy ↑ | 92.40±1.96 | 93.74±0.66 | 93.34±1.89 | 93.71±0.88 | 93.55±0.85 | 93.51±0.71 |
| | | | Equity Gap ↓ | 6.91±1.85 | 6.53±1.16 (5.50%) | 6.81±1.11 | 6.21±0.79 (8.81%) | 4.91±2.07 | 2.89±0.75 (41.14%) |
| Photo | 78.33±1.95 | 88.42±2.60 | Accuracy ↑ | 87.94±2.42 | 89.86±1.89 | 89.32±3.24 | 89.29±1.77 | 89.46±1.86 | 89.94±2.07 |
| | | | Equity Gap ↓ | 47.34±11.83 | 26.81±7.75 (43.37%) | 42.54±17.55 | 28.86±6.35 (32.16%) | 21.08±6.11 | 18.76±2.70 (11.01%) |
| Macro Avg. | **75.03** | **83.41** | Accuracy ↑ | **83.11** | **83.61** | **83.63** | **83.50** | **83.32** | **83.34** |
| | | | Equity Gap ↓ | **27.35** | **16.92** (38.14%) | **26.53** | **17.64** (33.51%) | **22.90** | **14.33** (37.42%) |

## H  Ablation under Transductive Setting

We further ablate the proposed objective under the transductive setting. Figure 7 shows the positive contribution of different components in the full `TED` objective. This indicates that teacher reliability weighting provides useful guidance for reducing class-conditional missed transfer.

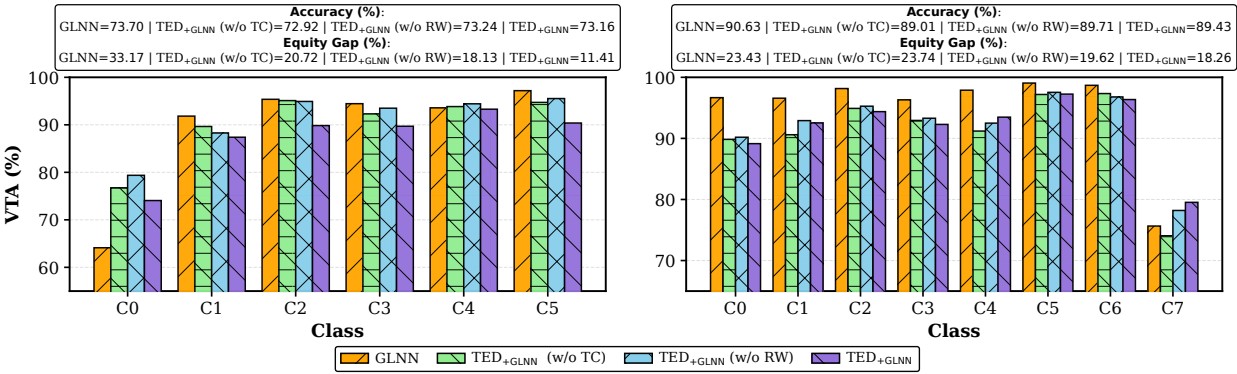

Figure 7: Ablation analysis of `TED` on CiteSeer (left) and Amazon-Photo (right) in transductive setting, show that removing reliability weighting or teacher-correctness filtering weakens the reduction in class-conditional transfer disparity. In both datasets, `TED` also improves the VTA of the most under-transferred class under vanilla GLNN

## I  Impact of Tolerance budget $\rho$ and Hyperparameter $\lambda_{\text{TED}}$

We analyze the sensitivity of `TED` to its main hyperparameters using GLNN as the base distillation method. The tolerance budget $\rho$ determines how much class-wise missed-transfer risk is allowed, while $\lambda_{\text{TED}}$ controls the strength of the transfer-equity regularizer. Since GLNN also includes the distillation weight $\alpha$, we additionally vary $\alpha$ to examine accuracy–equity tradeoff based on the base method's own hyperparameters.

Figure 8 compares $\rho = 0.05$ and $\rho = 0.1$ under the transductive setting. The results show that both choices maintain competitive accuracy, while the stricter tolerance $\rho = 0.05$ generally yields a lower Equity Gap. This is expected, since a smaller tolerance penalizes deviations in class-wise missed transfer more strongly. We use $\rho = 0.1$ in the main experiments as a conservative setting that reduces transfer disparity while avoiding an overly restrictive constraint.

Figure 9 reports sensitivity to $\lambda_{\text{TED}}$ and $\alpha$ on CiteSeer and Amazon-Photo under both transductive and inductive settings. The results show that nonzero values of $\lambda_{\text{TED}}$ consistently improve the accuracy–equity tradeoff relative to disabling the `TED` regularizer. Across datasets and settings, moderate values provide stable reductions in Equity Gap with limited changes in accuracy.

The sensitivity to $\alpha$ shows a similar pattern. Moderate reliability weighting improves Equity Gap while preserving predictive utility, whereas extreme values can degrade the tradeoff. This supports the design of `TED`: teacher reliability is useful for identifying valid transferable knowledge, but it should complement rather than dominate the distillation loss. Overall, the sensitivity analysis indicates that `TED` does not depend on a narrowly tuned hyperparameter choice and remains effective across a reasonable range of settings.

## J  Class-wise Transfer Profile

We use Equity Gap as the primary transfer-equity metric because it directly captures the largest class-conditional disparity in inherited correct teacher knowledge. Since Equity Gap is a range-based summary, we complement it with class-wise VTA profiles to evaluate the underlying per-label transfer behavior.

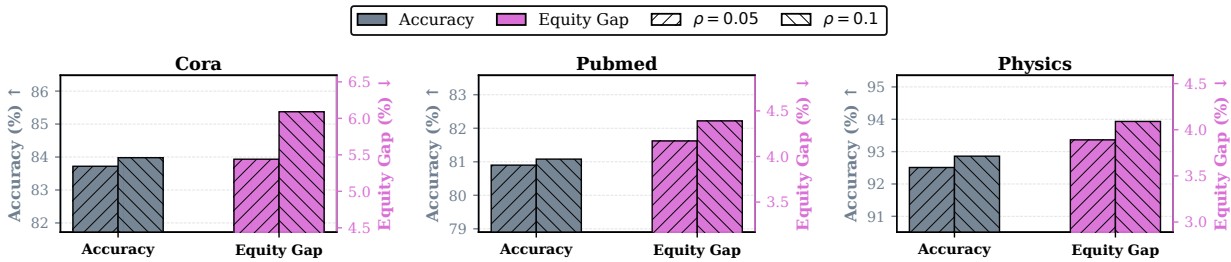

Figure 8: Sensitivity to the tolerance budget $\rho$ under the transductive setting.

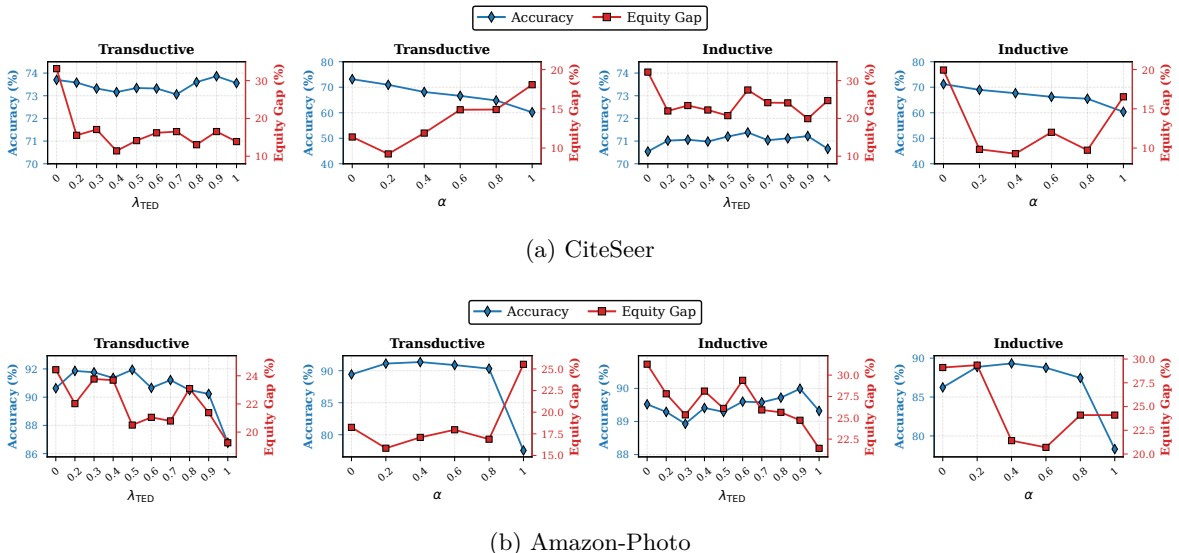

(a) CiteSeer

(b) Amazon-Photo

Figure 9: Impact of the TED regularization strength $\lambda_{\text{TED}}$ and the GLNN distillation weight $\alpha$ on accuracy and Equity Gap. Results are shown for CiteSeer and Amazon-Photo under both transductive and inductive settings. Moderate nonzero values of $\lambda_{\text{TED}}$ generally reduce Equity Gap while maintaining comparable accuracy, indicating that the transfer-equity regularizer improves class-wise transfer behavior without requiring a narrowly tuned value. The sensitivity to $\alpha$ shows the expected accuracy–equity tradeoff of the base distillation objective.

Table 6 provides a diagnostic view of class-conditional transfer beyond the scalar Equity Gap. For each valid class $\mathcal{C}_{\text{valid}} = \{c : |T_c| > 0\}$, VTA measures the fraction of teacher-correct examples whose teacher prediction is inherited by the student. We report four complementary VTA-based quantities. MeanVTA is the average class-wise VTA over valid classes,

$$\text{MeanVTA} = \frac{1}{|\mathcal{C}_{\text{valid}}|} \sum_{c \in \mathcal{C}_{\text{valid}}} \text{VTA}_c$$

WorstVTA is the lowest class-wise VTA,

$$\text{WorstVTA} = \min_{c \in \mathcal{C}_{\text{valid}}} \text{VTA}_c$$

and measures transfer to the least-inherited class. StdVTA is the standard deviation of class-wise VTA,

$$\text{StdVTA} = \sqrt{\frac{1}{|\mathcal{C}_{\text{valid}}|} \sum_{c \in \mathcal{C}_{\text{valid}}} \left( \text{VTA}_c - \text{MeanVTA} \right)^2}$$

Table 6: Transductive comparison of GLNN, Balanced KD, DRO, Focal Loss, and $\text{TED}_{+\text{GLNN}}$ across accuracy and class-wise VTA diagnostics. MeanVTA, WorstVTA, and StdVTA are computed over valid classes $\mathcal{C}_{\text{valid}}$. EG denotes the Equity Gap. All values are reported in percentage points as the mean over five independent runs. Best results are shown in **bold** and second-best results are underlined. The bottom block reports average dataset-wise ranks, where lower is better.

| Dataset | Metric | GLNN | Balanced KD | DRO | Focal Loss | TED |
|---|---|---|---|---|---|---|
| Cora | Acc.↑ | $83.68 \pm 0.91$ | $\underline{83.74 \pm 0.61}$ | $83.34 \pm 0.83$ | $83.28 \pm 0.69$ | $\mathbf{83.98 \pm 0.71}$ |
| | MeanVTA↑ | $95.61 \pm 0.80$ | $\underline{96.35 \pm 1.11}$ | $96.02 \pm 0.72$ | $95.28 \pm 0.46$ | $\mathbf{96.53 \pm 0.49}$ |
| | WorstVTA↑ | $90.26 \pm 1.88$ | $91.06 \pm 2.09$ | $\underline{91.21 \pm 1.51}$ | $89.62 \pm 1.79$ | $\mathbf{92.75 \pm 0.76}$ |
| | StdVTA↓ | $3.22 \pm 0.74$ | $2.88 \pm 0.65$ | $\underline{2.84 \pm 0.60}$ | $3.10 \pm 0.60$ | $\mathbf{1.97 \pm 0.21}$ |
| | EG↓ | $\underline{7.41 \pm 3.49}$ | $8.39 \pm 1.93$ | $8.12 \pm 1.50$ | $8.92 \pm 2.02$ | $\mathbf{6.09 \pm 0.69}$ |
| CiteSeer | Acc.↑ | $\mathbf{73.70 \pm 0.74}$ | $72.66 \pm 0.82$ | $72.32 \pm 0.61$ | $\underline{73.54 \pm 0.69}$ | $73.16 \pm 0.22$ |
| | MeanVTA↑ | $89.21 \pm 3.69$ | $82.70 \pm 5.77$ | $\underline{92.39 \pm 1.33}$ | $89.11 \pm 3.70$ | $\mathbf{92.51 \pm 0.89}$ |
| | WorstVTA↑ | $64.33 \pm 16.28$ | $31.89 \pm 28.30$ | $\mathbf{87.15 \pm 2.37}$ | $64.31 \pm 17.43$ | $\underline{86.44 \pm 1.56}$ |
| | StdVTA↓ | $11.36 \pm 5.56$ | $22.94 \pm 10.04$ | $\mathbf{3.65 \pm 0.61}$ | $11.49 \pm 5.87$ | $\underline{4.07 \pm 0.55}$ |
| | EG↓ | $33.17 \pm 21.14$ | $64.89 \pm 27.71$ | $\underline{11.65 \pm 2.14}$ | $32.99 \pm 16.57$ | $\mathbf{11.41 \pm 1.39}$ |
| PubMed | Acc.↑ | $79.86 \pm 0.70$ | $80.86 \pm 0.54$ | $80.40 \pm 0.78$ | $\underline{81.00 \pm 0.43}$ | $\mathbf{81.08 \pm 0.32}$ |
| | MeanVTA↑ | $93.67 \pm 0.57$ | $93.90 \pm 1.02$ | $93.94 \pm 1.21$ | $\underline{94.34 \pm 1.08}$ | $\mathbf{94.67 \pm 0.72}$ |
| | WorstVTA↑ | $89.82 \pm 0.75$ | $91.02 \pm 1.83$ | $91.19 \pm 2.41$ | $\underline{91.76 \pm 2.23}$ | $\mathbf{92.32 \pm 2.27}$ |
| | StdVTA↓ | $2.92 \pm 0.33$ | $2.58 \pm 0.72$ | $2.43 \pm 0.97$ | $\underline{2.15 \pm 1.01}$ | $\mathbf{2.03 \pm 0.97}$ |
| | EG↓ | $\underline{5.16 \pm 1.92}$ | $6.23 \pm 1.76$ | $5.82 \pm 2.42$ | $5.19 \pm 2.44$ | $\mathbf{4.39 \pm 2.33}$ |
| CS | Acc.↑ | $\mathbf{93.94 \pm 0.30}$ | $93.46 \pm 0.57$ | $92.41 \pm 1.03$ | $93.45 \pm 0.72$ | $\underline{93.66 \pm 0.26}$ |
| | MeanVTA↑ | $95.91 \pm 1.12$ | $96.17 \pm 1.21$ | $\mathbf{97.16 \pm 1.12}$ | $95.95 \pm 1.00$ | $\underline{96.28 \pm 0.70}$ |
| | WorstVTA↑ | $88.77 \pm 2.14$ | $89.81 \pm 2.41$ | $\mathbf{92.57 \pm 2.60}$ | $89.20 \pm 2.03$ | $\underline{90.12 \pm 1.76}$ |
| | StdVTA↓ | $3.18 \pm 0.49$ | $3.00 \pm 0.67$ | $\mathbf{2.02 \pm 0.88}$ | $3.15 \pm 0.45$ | $\underline{2.71 \pm 0.40}$ |
| | EG↓ | $9.63 \pm 2.14$ | $9.60 \pm 2.18$ | $\underline{9.01 \pm 2.60}$ | $10.22 \pm 1.80$ | $\mathbf{8.51 \pm 1.53}$ |
| Physics | Acc.↑ | $\mathbf{93.33 \pm 0.89}$ | $\underline{93.06 \pm 0.79}$ | $90.59 \pm 1.53$ | $\mathbf{93.33 \pm 0.93}$ | $92.86 \pm 0.93$ |
| | MeanVTA↑ | $96.44 \pm 1.20$ | $96.19 \pm 1.15$ | $\mathbf{97.18 \pm 0.82}$ | $96.65 \pm 1.35$ | $\underline{96.68 \pm 0.75}$ |
| | WorstVTA↑ | $92.76 \pm 2.83$ | $91.97 \pm 2.75$ | $\underline{93.47 \pm 2.31}$ | $92.86 \pm 2.89$ | $\mathbf{94.11 \pm 1.52}$ |
| | StdVTA↓ | $2.20 \pm 0.94$ | $2.44 \pm 0.92$ | $\underline{2.15 \pm 0.93}$ | $2.24 \pm 0.91$ | $\mathbf{1.55 \pm 0.48}$ |
| | EG↓ | $\underline{5.05 \pm 1.73}$ | $7.07 \pm 2.94$ | $5.85 \pm 2.49$ | $6.35 \pm 2.95$ | $\mathbf{4.09 \pm 1.46}$ |
| Amazon-Photo | Acc.↑ | $90.63 \pm 1.66$ | $\underline{91.91 \pm 1.65}$ | $81.22 \pm 1.68$ | $\mathbf{91.98 \pm 1.84}$ | $89.43 \pm 2.40$ |
| | MeanVTA↑ | $\underline{95.12 \pm 1.21}$ | $94.70 \pm 1.13$ | $86.81 \pm 0.85$ | $\mathbf{95.36 \pm 0.71}$ | $91.87 \pm 2.13$ |
| | WorstVTA↑ | $76.51 \pm 5.81$ | $73.51 \pm 7.31$ | $75.53 \pm 2.38$ | $\underline{77.74 \pm 5.07}$ | $\mathbf{79.37 \pm 5.14}$ |
| | StdVTA↓ | $7.17 \pm 1.81$ | $8.14 \pm 2.31$ | $\underline{6.42 \pm 0.93}$ | $6.85 \pm 1.65$ | $\mathbf{5.58 \pm 1.33}$ |
| | EG↓ | $23.43 \pm 7.78$ | $25.72 \pm 7.21$ | $\underline{19.47 \pm 2.36}$ | $21.60 \pm 4.97$ | $\mathbf{18.26 \pm 5.03}$ |
| Avg. rank | Acc. | $\mathbf{2.42}$ | $2.83$ | $4.67$ | $2.58$ | $\underline{2.50}$ |
| | MeanVTA | $3.83$ | $3.67$ | $\underline{2.50}$ | $3.17$ | $\mathbf{1.83}$ |
| | WorstVTA | $4.00$ | $4.17$ | $\underline{2.17}$ | $3.33$ | $\mathbf{1.33}$ |
| | StdVTA | $4.17$ | $4.17$ | $\underline{1.83}$ | $3.50$ | $\mathbf{1.33}$ |
| | EG | $3.00$ | $4.50$ | $\underline{2.67}$ | $3.83$ | $\mathbf{1.00}$ |

which captures dispersion in transfer reliability across labels. The results shows that TED provides the strongest overall transfer-reliability profile among the compared methods. TED obtains the best overall rank, substantially ahead of Focal Loss, DRO, GLNN, and Balanced KD. This indicates that TED is the most consistently effective method when predictive utility and class-wise transfer reliability are considered jointly. The advantage of TED is most pronounced on the transfer-disparity metrics that directly reflect the objective of this work. It achieves the best average rank for EG, StdVTA, and WorstVTA, showing that it consistently reduces the range of class-wise transfer outcomes, decreases dispersion across labels, and improves the least-transferred class. TED also obtains the best MeanVTA rank, indicating that the improvement in lower-tail transfer does not generally come at the cost of average valid teacher–student agreement. For accuracy, TED ranks second just behind the vanilla GLNN, confirming that the transfer-reliability gains are achieved while preserving competitive predictive utility. The comparison also clarifies why generic reweighting and robust-optimization baselines are insufficient. Balanced KD does not consistently reduce transfer disparity and has the weakest overall average rank. Focal Loss is competitive on accuracy but is less effective on EG,

WorstVTA, and StdVTA, suggesting that student-side hard-example emphasis does not directly address class-conditional missed transfer. DRO improves some transfer metrics, but it is less stable overall and has a substantially weaker accuracy rank. In contrast, `TED` is the only method that is consistently top-ranked on the transfer-specific metrics while remaining competitive on accuracy. This supports the need for a reliable teacher-transfer objective rather than generic frequency-, hardness-, or worst-loss-based reweighting.

Overall, these results indicate that `TED` improves class-conditional transfer reliability in a comprehensive sense, not merely through a favorable change in a single summary metric. It achieves the strongest overall tradeoff across predictive accuracy, worst-class inheritance, and dispersion of class-wise transfer, supporting its role as a transfer-specific objective rather than a generic reweighting heuristic.

**Class-wise Distribution of Teacher-Correct Samples**  Table 7 reports the teacher-correct support used to compute class-wise VTA under the transductive setting for a single trial. All homophilic datasets have nonzero support for every class, so VTA is well-defined across labels.

Table 7: Teacher-correct support under the transductive setting. For each dataset, we report the total number of evaluation samples correctly classified by the teacher and the corresponding class-wise teacher-correct counts. These counts define the valid support used to compute class-wise VTA.

| Dataset | # Classes | Correct | Min | Median | Class-wise teacher-correct counts |
|---|---|---|---|---|---|
| Cora | 7 | 816 | 49 | 100 | 100, 76, 125, 258, 128, 80, 49 |
| CiteSeer | 6 | 774 | 81 | 130.5 | 81, 106, 131, 180, 146, 130 |
| PubMed | 3 | 765 | 142 | 279 | 142, 344, 279 |
| Coauthor CS | 15 | 15,689 | 61 | 746 | 577, 202, 1,764, 326, 1,160, 1,961, 279, 797, 510, 61, 1,290, 1,735, 341, 3,940, 746 |
| Coauthor Physics | 5 | 29,767 | 2,163 | 4,430 | 5,539, 4,430, 15,319, 2,316, 2,163 |
| Amazon Photo | 8 | 6,454 | 233 | 733 | 300, 1,365, 615, 789, 723, 743, 1,686, 233 |
| Wisconsin | 5 | 46 | 1 | 7 | 1,16,21,7,1 |
| Cornell | 5 | 33 | 2 | - | 4, N/A, 2,20,7 |

# K   Preserving Predictive Utility Beyond Accuracy

Table 8 evaluates whether the reduction in class-conditional transfer disparity comes at the expense of standard predictive metrics. We report accuracy as the average node classification performance, macro-F1 as a class-balanced measure of predictive quality, and minority-class recall as the recall of the least frequent class in each dataset. These metrics complement EG by testing whether reducing missed transfer harms overall prediction, balanced class performance, or the least frequent class.

The results show that `TED` preserves competitive predictive utility while reducing transfer disparity. In the average-rank summary, `TED` obtains the best macro-F1 rank, indicating that the improvement in EG is not obtained by sacrificing overall accuracy or balanced predictive performance. At the same time, `TED` achieves the best EG rank, reducing class-conditional transfer disparity more consistently than the reweighting and robust-optimization baselines.

The minority-class recall results further show that `TED` does not systematically harm the least frequent class. `TED` obtains the second-best average rank for minority recall, behind DRO. However, DRO's strong minority recall comes with a substantial loss in accuracy and macro-F1 on several datasets, especially CS and Amazon-Photo. In contrast, `TED` provides a more balanced tradeoff: it remains competitive on minority recall while achieving the best EG and macro-F1 ranks and maintaining top-ranked accuracy. For example, on CiteSeer, `TED` improves minority recall from 30.39% to 42.34%, improves macro-F1 from 69.66% to 70.00%, and reduces EG from 33.17% to 11.41%. On Amazon-Photo, `TED` improves minority recall from 68.26% to 73.74% and reduces EG from 23.43% to 18.26%.

Overall, these results address the concern that `TED` might reduce transfer disparity by degrading conventional predictive metrics. `TED` achieves the strongest reduction in EG while remaining competitive in accuracy, obtaining the best macro-F1 rank, and maintaining strong minority-class recall. This supports the claim that `TED` improves class-conditional transfer reliability without heavily trading away the predictive utility of graph-free distillation.

Table 8: Transductive comparison of GLNN, Balanced KD, DRO, Focal Loss, and TED$_{+GLNN}$ across accuracy, macro F1, minority-class recall, and Equity Gap. Minority-class recall is computed for the least frequent test class in each dataset. All values are reported in percentage points as the mean ± standard deviation over five independent runs. Best results are shown in **bold**, and second-best results are underlined. The bottom block reports average dataset-wise ranks, where lower is better.

| Dataset | Metric | GLNN | Balanced KD | DRO | Focal Loss | TED |
|---|---|---|---|---|---|---|
| Cora | Acc.↑ | $83.68 \pm 0.91$ | $\underline{83.74 \pm 0.61}$ | $83.34 \pm 0.83$ | $83.28 \pm 0.69$ | $\mathbf{83.98 \pm 0.71}$ |
| | Macro F1↑ | $\underline{82.44 \pm 0.61}$ | $82.32 \pm 0.68$ | $82.14 \pm 0.83$ | $81.93 \pm 0.76$ | $\mathbf{82.69 \pm 0.66}$ |
| | Minority Recall↑ | $\underline{83.44 \pm 6.75}$ | $83.12 \pm 6.73$ | $\underline{83.44 \pm 5.90}$ | $81.25 \pm 5.41$ | $\mathbf{83.75 \pm 5.00}$ |
| | EG↓ | $\underline{7.41 \pm 3.49}$ | $8.39 \pm 1.93$ | $8.12 \pm 1.50$ | $8.92 \pm 2.02$ | $\mathbf{6.09 \pm 0.69}$ |
| CiteSeer | Acc.↑ | $\mathbf{73.70 \pm 0.74}$ | $72.66 \pm 0.82$ | $72.32 \pm 0.61$ | $\underline{73.54 \pm 0.69}$ | $73.16 \pm 0.22$ |
| | Macro F1↑ | $\underline{69.66 \pm 0.70}$ | $66.27 \pm 2.74$ | $69.47 \pm 0.47$ | $69.36 \pm 0.95$ | $\mathbf{70.00 \pm 0.23}$ |
| | Minority Recall↑ | $30.39 \pm 7.37$ | $15.58 \pm 12.27$ | $\mathbf{44.16 \pm 1.84}$ | $30.13 \pm 8.23$ | $\underline{42.34 \pm 2.92}$ |
| | EG↓ | $33.17 \pm 21.14$ | $64.89 \pm 27.71$ | $\underline{11.65 \pm 2.14}$ | $32.99 \pm 16.57$ | $\mathbf{11.41 \pm 1.39}$ |
| PubMed | Acc.↑ | $79.86 \pm 0.70$ | $80.86 \pm 0.54$ | $80.40 \pm 0.78$ | $\underline{81.00 \pm 0.43}$ | $\mathbf{81.08 \pm 0.32}$ |
| | Macro F1↑ | $80.65 \pm 0.21$ | $80.77 \pm 0.52$ | $80.33 \pm 0.87$ | $\underline{80.92 \pm 0.38}$ | $\mathbf{81.04 \pm 0.34}$ |
| | Minority Recall↑ | $79.44 \pm 1.11$ | $79.56 \pm 0.65$ | $\mathbf{82.11 \pm 0.96}$ | $79.67 \pm 2.80$ | $\underline{80.44 \pm 0.65}$ |
| | EG↓ | $\underline{5.16 \pm 1.92}$ | $6.23 \pm 1.76$ | $5.82 \pm 2.42$ | $5.19 \pm 2.44$ | $\mathbf{4.39 \pm 2.33}$ |
| CS | Acc.↑ | $\mathbf{93.94 \pm 0.30}$ | $93.46 \pm 0.57$ | $92.41 \pm 1.03$ | $93.45 \pm 0.72$ | $\underline{93.66 \pm 0.26}$ |
| | Macro F1↑ | $90.61 \pm 0.90$ | $\underline{90.84 \pm 0.72}$ | $90.32 \pm 1.30$ | $90.83 \pm 0.81$ | $\mathbf{91.18 \pm 0.30}$ |
| | Minority Recall↑ | $89.41 \pm 4.87$ | $88.82 \pm 5.63$ | $\mathbf{98.24 \pm 2.85}$ | $88.24 \pm 4.92$ | $\underline{89.71 \pm 3.95}$ |
| | EG↓ | $9.63 \pm 2.14$ | $9.60 \pm 2.18$ | $\underline{9.01 \pm 2.60}$ | $10.22 \pm 1.80$ | $\mathbf{8.51 \pm 1.53}$ |
| Physics | Acc.↑ | $\mathbf{93.33 \pm 0.89}$ | $\underline{93.06 \pm 0.79}$ | $90.59 \pm 1.53$ | $\mathbf{93.33 \pm 0.93}$ | $92.86 \pm 0.93$ |
| | Macro F1↑ | $\underline{91.20 \pm 1.40}$ | $90.89 \pm 1.31$ | $88.79 \pm 1.64$ | $\mathbf{91.23 \pm 1.43}$ | $90.68 \pm 1.38$ |
| | Minority Recall↑ | $93.35 \pm 0.93$ | $93.39 \pm 0.99$ | $\mathbf{95.02 \pm 1.69}$ | $93.44 \pm 1.04$ | $\underline{94.95 \pm 1.51}$ |
| | EG↓ | $\underline{5.05 \pm 1.73}$ | $7.07 \pm 2.94$ | $5.85 \pm 2.49$ | $6.35 \pm 2.95$ | $\mathbf{4.09 \pm 1.46}$ |
| Amazon-Photo | Acc.↑ | $90.63 \pm 1.66$ | $\underline{91.91 \pm 1.65}$ | $81.22 \pm 1.68$ | $\mathbf{91.98 \pm 1.84}$ | $89.43 \pm 2.40$ |
| | Macro F1↑ | $\mathbf{90.38 \pm 1.06}$ | $90.20 \pm 1.19$ | $79.85 \pm 1.33$ | $\underline{90.34 \pm 1.46}$ | $87.43 \pm 2.37$ |
| | Minority Recall↑ | $68.26 \pm 7.56$ | $65.41 \pm 7.55$ | $\mathbf{84.41 \pm 5.16}$ | $68.68 \pm 6.80$ | $\underline{73.74 \pm 5.65}$ |
| | EG↓ | $23.43 \pm 7.78$ | $25.72 \pm 7.21$ | $\underline{19.47 \pm 2.36}$ | $21.60 \pm 4.97$ | $\mathbf{18.26 \pm 5.03}$ |
| Avg. rank | Acc. | $\mathbf{2.42}$ | $2.83$ | $4.67$ | $2.58$ | $\underline{2.50}$ |
| | Macro F1 | $\underline{2.50}$ | $3.17$ | $4.50$ | $2.83$ | $\mathbf{2.00}$ |
| | Minority Recall | $3.75$ | $4.33$ | $\mathbf{1.25}$ | $3.83$ | $\underline{1.83}$ |
| | EG | $3.00$ | $4.50$ | $\underline{2.67}$ | $3.83$ | $\mathbf{1.00}$ |

## L  Visualization of Representations

Beyond accuracy and Equity Gap, we visualize the penultimate-layer student embeddings with t-SNE (Van der Maaten & Hinton, 2008) to qualitatively examine the learned representation geometry. Figures 10 and 11 show t-SNE visualizations of the penultimate-layer student representations on Cora and CiteSeer, respectively. Figure 12 shows results on Amazon-Photo for GLNN, KRD, FAITH, and their TED-augmented variants. These plots provide a qualitative view of the representation geometry learned by the base graph-free students and their TED-augmented variants. Across both datasets, the TED variants show slightly more class-consistent structure, with several classes appearing more compact.

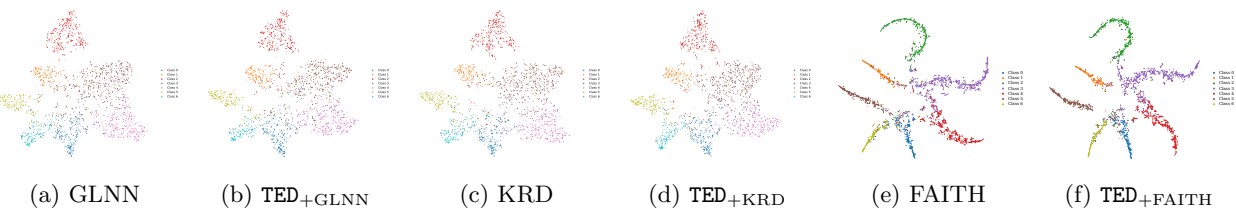

(a) GLNN  (b) TED$_{+GLNN}$  (c) KRD  (d) TED$_{+KRD}$  (e) FAITH  (f) TED$_{+FAITH}$

Figure 10: t-SNE visualization of learned representations on Cora; colors denote different classes.

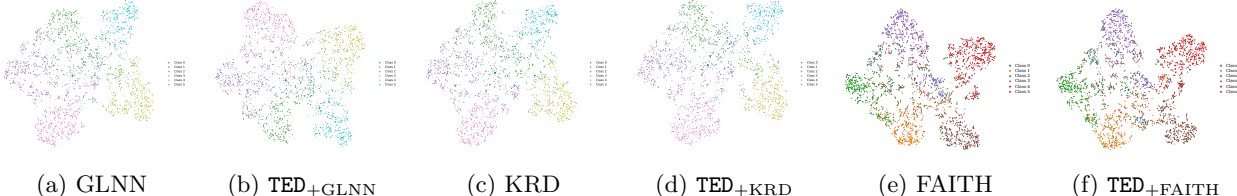

(a) GLNN     (b) TED$_{+\text{GLNN}}$     (c) KRD     (d) TED$_{+\text{KRD}}$     (e) FAITH     (f) TED$_{+\text{FAITH}}$

Figure 11: t-SNE visualization of learned representations on `Citeseer`; colors denote different classes.

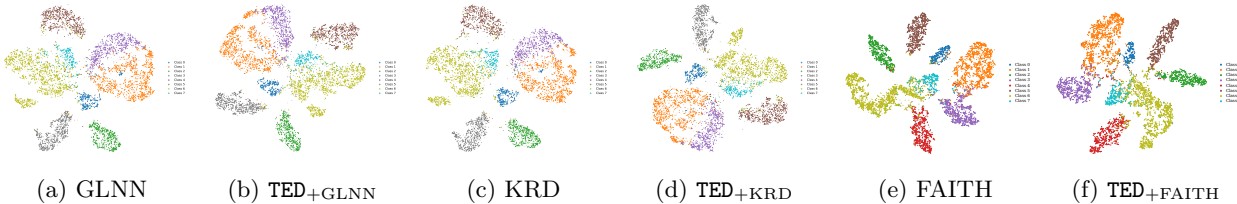

(a) GLNN     (b) TED$_{+\text{GLNN}}$     (c) KRD     (d) TED$_{+\text{KRD}}$     (e) FAITH     (f) TED$_{+\text{FAITH}}$

Figure 12: t-SNE visualization of learned student representations on Amazon-Photo.

## M   Limitations and Future Work

VTA is an offline, label-dependent diagnostic for evaluating valid teacher-knowledge inheritance. It is not intended as a label-free runtime monitoring measure. Computing VTA requires ground-truth labels in order to identify the teacher-correct set $T_c = \{v \in V_c : \hat{y}_v^t = y_v\}$, and therefore requires a labeled test set. VTA should thus be viewed as an evaluation and auditing metric for graph-free distillation rather than as a deployment-time signal available on unlabeled data.

TED uses teacher correctness on labeled nodes to identify reliable teacher decisions. In this work, we assume that teacher confidence reliably reflects predictive reliability. However, when each class has only a very small number of labeled examples, the reliability signal can become noisy: teacher correctness may be estimated from too few nodes to represent the class-wise behavior of the teacher reliably. We therefore include an additional low-label sensitivity analysis to examine how TED behaves under reduced labeled support. We vary the number of labeled nodes per class using $k$-shot settings. In particular, we evaluate $k = 1$ and $k = 5$ labeled nodes per class on CiteSeer and Amazon-Photo datasets under transductive setting. We compare the base GLNN model with TED$_{+\text{GLNN}}$ and report Accuracy, Equity Gap, MeanVTA, WorstVTA, and StdVTA. These metrics allow us to evaluate both predictive utility and the stability of class-conditional transfer under scarce labeled support.

Figure 13 shows that TED continues to reduce class-conditional transfer disparity even under severely reduced labeled support. In the extreme 1-shot setting, where the reliability signal is estimated from only one labeled example per class, TED still reduces Equity Gap on both datasets. On CiteSeer, EG decreases from 98.50% to 95.36% with no change in accuracy. On Amazon-Photo, EG decreases from 62.78% to 60.06%, with a slight accuracy reduction. The benefits become clearer in the 5-shot setting, where the reliability signal is less noisy. On CiteSeer, TED improves accuracy from 59.80% to 60.73% while reducing EG substantially from 87.14% to 58.41%. On Amazon-Photo, TED reduces EG from 42.40% to 37.47%, with only a small accuracy change from 88.03% to 87.48%. Thus, with modest labeled support, TED provides a stronger and more stable reduction in class-conditional missed transfer.

This analysis clarifies the scope of TED. The method is most suitable for supervised or semi-supervised GNN-to-MLP distillation settings with nontrivial labeled support per class. Under extremely scarce-label regimes, the teacher-correctness signal can have high variance, which may weaken the class-wise missed-transfer estimate.

Future work can therefore estimate teacher reliability from complementary label-efficient signals, such as prediction stability under perturbations, agreement among multiple teacher models, or confidence calibra-

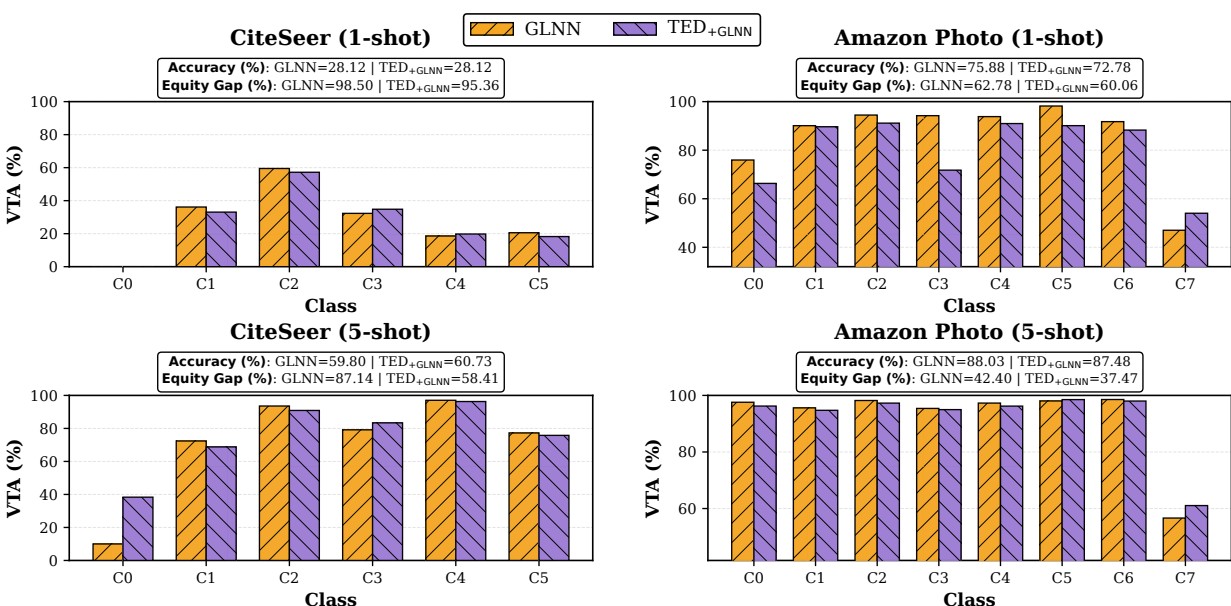

Figure 13: Low-label sensitivity of TED. We evaluate TED$_{+\text{GLNN}}$ in 1-shot and 5-shot settings on CiteSeer and Amazon-Photo under transductive setting.

tion. Another promising direction is semi-supervised reliability estimation, where unlabeled nodes contribute through calibrated pseudo-labels or consistency-based teacher confidence. These extensions would retain the transfer-specific objective of TED while improving its robustness when labeled support per class is extremely limited.is to make reliability estimation less dependent on scarce labeled support.

**Broader Impact.** Graph-free GNN-to-MLP distillation is often motivated by deployment efficiency, since the student avoids inference-time message passing. Our results show that this efficiency can hide class-conditional transfer failures: a student may maintain strong average accuracy while failing to inherit valid teacher knowledge for particular labels. In real node-classification systems, such missed transfer could be especially important when the affected labels are rare, minority, or high-risk classes. VTA and Equity Gap therefore provide an offline, label-dependent pre-deployment auditing tool for detecting class-wise transfer failures before releasing a graph-free student. These diagnostics should be interpreted as measures of class-conditional transfer reliability, not as demographic fairness guarantees.

