# OpenReview forum: "Do MLPs Inherit GNN Knowledge Uniformly? Class-Conditional Transfer Equity in Graph-Free Distillation"
_TMLR — Under review for TMLR_

### Review · Reviewer_cbLX · 2026-07-01

**Summary Of Contributions:**

This paper studies class-conditional transfer imbalance in GNN-to-MLP knowledge distillation, arguing that standard evaluation using average accuracy can obscure significant variation in how effectively different classes inherit teacher knowledge. To address this issue, the authors propose VTA (Valid Transfer Agreement) to measure class-wise preservation of teacher decisions and define Equity Gap to quantify disparities across classes. The authors introduce TED (Transfer Equity Distillation), a training-time regularization method that encourages more balanced transfer by penalizing classes with insufficient retention of reliable teacher knowledge. Experiments across multiple datasets and distillation frameworks show that TED consistently reduces Equity Gap while maintaining competitive overall accuracy.

**Additional Comments:**

I would recommend a major revision.

**Audience:**

Yes

**Audience Explanation:**

•	The paper highlights a practical and meaningful issue: average accuracy can mask significant class-level disparities in knowledge distillation performance, which is particularly relevant in deployment settings.
•	The proposed VTA metric is well-motivated and helps isolate student failure from teacher misclassification by conditioning on correct teacher predictions.
•	TED is a plug-and-play method that does not modify model architecture or inference procedure, making it easy to integrate into existing pipelines.
•	The experimental evaluation is fairly comprehensive, covering multiple datasets, different distillation frameworks, and both homophilic and heterophilic graph settings, with generally consistent results.

**Broader Impact Concerns:**

The proposed TED method appears closely related to existing class-wise reweighting strategies combined with confidence-based filtering, and is conceptually similar to approaches used in long-tailed learning, such as focal loss and class-balanced loss, as well as prior reweighting-based knowledge distillation methods. The paper does not clearly articulate what fundamentally distinguishes TED from these existing techniques.

**Claims And Evidence:**

No

**Claims Explanation:**

The overall contribution appears closer to a class-balanced reweighting strategy adapted to the distillation setting, rather than a fundamentally novel methodological framework. The theoretical analysis is relatively weak, and the empirical study, while extensive, lacks deeper mechanistic explanations. The explanations should be significantly expanded.

**Requested Changes:**

•	The paper focuses primarily on identifying and mitigating the observed imbalance phenomenon, but provides limited explanation of why certain classes are more prone to transfer failure. Potential factors such as feature separability, graph homophily, or class imbalance are not systematically analyzed, making the proposed method appear more like an empirical correction rather than a principled, mechanism-driven approach.
•	The experimental evaluation lacks comparisons against stronger and more relevant baselines from long-tailed learning and robust optimization literature, such as focal loss, class-balanced loss, or distributionally robust optimization (DRO). Given that TED is essentially a reweighting-based method, the absence of these comparisons weakens the evidence for its effectiveness.

---

> ### Author Response · Authors · 2026-07-19
> **Response to Review: Part 1**
>
> We sincerely thank the reviewer for the careful reading and constructive feedback. We also appreciate the reviewer’s positive assessment that the paper studies a practical and meaningful issue: average accuracy can mask class-level disparities in GNN-to-MLP distillation, VTA is well-motivated because it isolates student failure from teacher misclassification, TED is plug-and-play without changing the architecture or inference procedure, and the experimental evaluation covers multiple datasets, distillation frameworks, and graph regimes.
>
> Below, we provide detailed point-by-point responses to the reviewer’s requested changes. All corresponding revisions are highlighted in yellow in the updated manuscript.
>
> ---
>
> #### **Change 1: TED objective and distinction from reweighting and DRO baselines**
>
> **Response:**
> We thank the reviewer for this important feedback. TED is different from class-frequency reweighting method. It regularizes a transfer-specific residual: the amount of correct and confident teacher knowledge that is not inherited by the graph-free student. TED is formulated as a bounded class-conditional missed-transfer risk. First, for each labeled training node $v$, the frozen teacher GNN produces a probability vector $p_v^t$ and a predicted label $\hat{y}_v^t$. The student MLP produces $p_v^s$. Let $y_v$ be the ground-truth label, $V_c$ be the set of labeled training nodes from class $c$, and $q_v^t \in [0,1]$ be the normalized confidence of the teacher prediction. TED defines the reliability weight
>
> $w_v^t = \mathbf{1}[\hat{y}_v^t = y_v] q_v^t .$
>
> Thus, an incorrect teacher prediction receives zero weight, while a correct but uncertain teacher prediction is downweighted. TED then defines a teacher-targeted transfer loss
>
> $\ell_v^{tr}(\phi) = -\log p^s_{v,\hat{y}_v^t},$
>
> where $p^s_{v,\hat{y}_v^t}$ is the probability assigned by the student to the teacher’s predicted class. This loss is large when the student fails to assign high probability to the teacher decision being transferred.
>
> For each class $c$, TED aggregates this quantity into the class-wise missed reliable transfer risk
>
> $R_c(\phi)=
> \frac{\sum_{v \in V_c} w_v^t \ell_v^{tr}(\phi)}
> {\sum_{v \in V_c} w_v^t}.$
>
> $R_c(\phi)$ measures how much correct and confident teacher knowledge for class $c$ is not inherited by the graph-free student.
>
> This is the key distinction from class-balanced KD, focal loss, and DRO: those methods operate on label frequency, student-side example difficulty, or generic worst-class loss, whereas $R_c(\phi)$ is a teacher-conditioned transfer residual aligned with VTA.
>
> This distinguishes TED from the baselines in **Table 2**. Class-Balanced KD uses label frequency, Focal Loss emphasizes examples that are difficult under the student’s prediction, and DRO optimizes worst-class risk. TED instead regularizes **missed reliable transfer conditioned on teacher correctness and confidence**, which is the failure mode measured by VTA and EG.
>
> To make this distinction empirical, the revised manuscript adds **Balanced KD**, **Focal Loss**, and **DRO** comparisons in **Table 2**. The results show that reweighting or robust optimization does not consistently match TED’s accuracy-EG tradeoff. In the macro-average comparison, Balanced KD increases EG from **13.98% to 20.32%**, while macro-average accuracy changes from **85.86% to 85.95%**. Focal Loss increases EG slightly from **13.98% to 14.21%**, while macro-average accuracy changes from **85.86% to 86.10%**. DRO reduces EG from **13.98% to 9.99%**, but with a larger accuracy drop from **85.86% to 83.38%**. TED gives the strongest tradeoff: it reduces macro-average EG from **13.98% to 8.79%**, corresponding to a **37.12% reduction** relative to GLNN, while macro-average accuracy changes from **85.86% to 85.70%**, corresponding to only a **0.19% reduction**.
>
> Furthermore, we have expanded the Related Work section to discuss class-balanced, focal-style, and distributionally robust objectives more explicitly and to clarify how TED differs from these approaches.
>
> > **Changes in the manuscript:**
> >
> > 1. Clarified the TED objective as class-conditional missed reliable transfer.
> > 2. Added **Balanced KD**, **Focal Loss**, and **DRO** comparisons in **Table 2**.
> > 3. Revised the related-work and experimental discussion to distinguish TED from label-frequency reweighting, focal loss, and robust optimization.

---

> > ### Author Response · Authors · 2026-07-19
> > **Response to Review: Part 2**
> >
> > #### **Change 2: Mechanistic explanation of why some classes are more prone to transfer failure**
> >
> > **Response:**
> > We thank the reviewer for this suggestion. To address this, the revised manuscript adds a new diagnostic analysis in **RQ6: Figure 6**. Specifically, we analyze three class-level factors:
> >
> > 1. **Class support:** the fraction of test samples belonging to each class.
> > 2. **Class-wise edge homophily:** the extent to which a class is connected to same-label neighbors.
> > 3. **Raw-feature separability:** a centroid-based separability score measuring how well a class is separated in the original feature space.
> >
> > **Figure 6** highlights the lowest-VTA class under vanilla GLNN on CiteSeer and Amazon-Photo. On both datasets, the lowest-VTA class has low class support, low edge homophily, and weak feature separability. This analysis also provides evidence that class-balanced reweighting alone is insufficient. If low-VTA classes were difficult only because of class frequency, then class-balanced KD should largely close the EG gap. The diagnostics suggest that transfer disparity arises from the combined effect of support, graph homophily, and feature separability. TED is better matched to this setting because it directly regularizes class-wise missed transfer of reliable teacher decisions, rather than assuming that frequency alone explains the failure.
> >
> > > **Changes in the manuscript:**
> > >
> > > 1. Added **RQ6: Diagnosing Class-Wise Transfer Disparities**.
> > > 2. Added **Figure 6**, analyzing class support, class-wise edge homophily, and feature separability.
> > > 3. Added discussion showing that low-VTA classes are associated with weaker graph-free recovery conditions.
> > > 4. Clarified why class-balanced reweighting alone is insufficient to explain or mitigate the observed transfer disparity.

---

### Review · Reviewer_KPFC · 2026-07-08

**Summary Of Contributions:**

This paper studies an important but often hidden failure mode in graph-free GNN-to-MLP distillation: whether an MLP student inherits the teacher GNN's knowledge uniformly across classes. The authors introduce Valid Transfer Agreement (VTA), a label-level metric intended to measure whether the student preserves teacher decisions on test samples where the teacher is correct. They further propose Transfer Equity Distillation (TED), a reliability-guided objective augmentation that aims to reduce class-conditional missed transfer without changing the teacher, student architecture, or graph-free inference procedure. According to the abstract, TED reduces class-level transfer disparity and improves worst-label teacher-student agreement across eight homophilic and heterophilic graph benchmarks while maintaining competitive accuracy and inference efficiency.

**Strengths.** The paper targets a meaningful evaluation gap: average accuracy can hide severe class-level failures in distilled graph-free models. The proposed VTA metric is intuitive and could complement standard accuracy and fidelity metrics. If TED can indeed be applied as a lightweight augmentation to existing GNN-to-MLP distillation methods, it would be practically useful.

**Weaknesses.** The currently visible material only includes the abstract and the OpenReview review form, not the full method, experimental setup, baselines, ablations, or quantitative tables. Therefore, I cannot fully assess whether the main claims are supported by accurate, clear, and convincing evidence. In particular, the formulation of TED, its distinction from reweighting / focal-style objectives / class-balanced distillation, and its actual benefits for minority or hard classes cannot be verified from the available material.

**Additional Comments:**

The topic is valuable and the problem statement is clear. However, based on the currently visible material, I cannot verify whether the method and experiments are sufficient to support acceptance. My view is that the work could meet TMLR's interest criterion if the full paper provides strong per-class evidence, strong baselines, and careful ablation studies. Without such evidence, the main empirical claims remain insufficiently supported.

**Audience:**

Yes

**Audience Explanation:**

The problem is relevant to graph learning, knowledge distillation, model compression, and robust/fair model evaluation. Researchers interested in graph-free inference, GNN deployment, distillation diagnostics, or class-conditional model behavior would likely find the findings useful. Even if the method is relatively simple, the observation that average distillation performance can hide class-conditional transfer failures would be worth knowing if supported by strong evidence.

**Broader Impact Concerns:**

I do not see major ethical concerns that would require a separate in-depth discussion. One point worth mentioning is that, if graph-free distillation is deployed in real node classification systems, class-conditional missed transfer may disproportionately harm minority or high-risk classes. The proposed class-level diagnostic metric could therefore also serve as a pre-deployment auditing tool. If the paper does not already include a broader impact statement, I recommend adding this point.

**Claims And Evidence:**

No

**Claims Explanation:**

The claims made in the abstract are plausible and potentially valuable, but they cannot be verified from the paper. The authors claim that TED reduces class-level transfer disparity, improves worst-label agreement, and preserves accuracy and graph-free inference efficiency across eight benchmarks. These are empirical claims that require complete experimental results, variance or confidence intervals, strong baselines, ablations, and failure-case analysis. Since the current material does not show this evidence, I cannot conclude that the claims are sufficiently supported.

To meet the TMLR evidence standard, the authors should at least provide or clarify:

1. The relationship between VTA and existing fidelity, teacher-student agreement, per-class accuracy / recall, and calibration-aware metrics, including what failure modes VTA captures that existing metrics miss.
2. The complete TED objective, the differentiable approximation used, additional hyperparameters, and why TED should be preferred over simple class reweighting, confidence filtering, focal loss, balanced KD, or related alternatives.
3. Per-dataset and per-class results for VTA, worst-label agreement, and average accuracy, ideally reported over multiple random seeds with standard deviations.
4. Ablations removing teacher-correctness filtering, confidence thresholding, and hardness weighting.
5. Sensitivity analysis with respect to teacher errors, teacher confidence miscalibration, class imbalance, and varying levels of heterophily.

**Requested Changes:**

1. Provide complete experimental results and statistical evidence showing that TED improves worst-label agreement and transfer disparity, rather than only average trends.
2. Include strong baselines, such as class-balanced KD, confidence-thresholded KD, focal-style KD, per-class reweighting, and calibration or reweighting methods that do not change the model architecture.
3. Clearly define VTA, its motivation, and its limitations, especially when the teacher has low accuracy or poor confidence calibration for some classes.
4. Demonstrate that TED's gains are not obtained at the expense of overall accuracy, macro-F1, minority-class recall, or calibration.

---

> ### Author Response · Authors · 2026-07-19
> **Response to Review: Part 1**
>
> We sincerely thank the reviewer for the thoughtful and constructive feedback. We appreciate that the reviewer recognizes the importance of the problem: average accuracy can hide severe class-level failures in graph-free GNN-to-MLP distillation, and VTA can provide a useful diagnostic for this hidden failure mode.
>
>
> Below, we provide detailed point-by-point responses to the reviewer’s requested changes. All corresponding revisions are highlighted in yellow in the updated manuscript.
>
> ---
>
> #### **Change 1: Relationship between VTA and existing metrics**
>
> **Response:**
> We thank the reviewer for this important feedback. The revised manuscript makes clearer that VTA is not intended to replace accuracy, fidelity, recall, or calibration metrics. Instead, it isolates a specific failure mode in graph-free distillation: whether a graph-free student preserves teacher decisions **on samples where the teacher is correct**.
>
> Accuracy measures whether the student predicts the ground-truth label, but it does not reveal whether the student inherited the graph-aware teacher’s correct decision. Raw teacher-student agreement or fidelity measures agreement with the teacher on all samples, but it also rewards imitation of teacher mistakes. VTA differs from these metrics by conditioning on teacher correctness and measures among examples where the graph-aware teacher is correct, how often does the graph-free student preserve that decision for each class?
>
> We also discuss the limitations of VTA in Appendix M.
>
> ---
>
> #### **Change 2: Ablations of TED**
>
> **Response:**
> We thank the reviewer for this important feedback. The revised manuscript includes these ablations in **Figure 5**.
>
> Starting from TED+GLNN, we evaluate two simplified variants:
>
> 1. **TED without teacher-correctness filtering (w/o TC):** incorrect teacher predictions are no longer excluded from the missed-transfer estimate.
> 2. **TED without reliability weighting (w/o RW):** teacher confidence weighting is removed.
>
> The results show that each component contributes positively.
>
>
> > **Changes in the manuscript:**
> >
> > 1. Added the w/o TC and w/o RW ablations in **Figure 5**.
>
> ---
>
> #### **Change 3: Utility metrics beyond accuracy**
>
> **Response:**
> We thank the reviewer for this suggestion. In the revised manuscript, we added **Appendix K: Table 8**, which reports **Accuracy**, **Macro-F1**, **Minority-class Recall**, and **EG** for GLNN, Balanced KD, DRO, Focal Loss, and TED.
>
> The average-rank summary in Table 8 shows that TED obtains the most favourable tradeoff between utility and disparity metrics.
>
> > **Changes in the manuscript:**
> >
> > 1. Added **Appendix K: Table 8** with Accuracy, Macro-F1, Minority-class Recall, and EG.
> > 2. Clarified that TED preserves competitive predictive utility while reducing transfer disparity.
>
> ---
>
> #### **Change 4: Broader impact and deployment interpretation**
>
> **Response:**
> We appreciate the reviewer’s suggestion that class-conditional missed transfer can matter in deployed node-classification systems. We have strengthened this discussion in the revised manuscript.
>
> First, in the **Introduction**, we emphasize the deployment interpretation of the problem: a graph-free MLP student may achieve strong average accuracy while still failing to preserve valid GNN teacher knowledge for particular classes. We explicitly note that this issue is relevant to deployed graph-based systems, where a fast graph-free student may appear accurate on average but behave inconsistently across categories.
>
> Finally, we added a dedicated **Broader Impact** paragraph in **Appendix M**. There, we state that graph-free distillation is motivated by deployment efficiency, but this efficiency can hide class-conditional transfer failures. We further clarify that VTA and EG can serve as offline, label-dependent pre-deployment auditing tools for detecting class-wise transfer failures before releasing a graph-free student.
>
>
> > **Changes in the manuscript:**
> >
> > 1. Strengthened the deployment-reliability discussion in the **Introduction**.
> > 2. Added a dedicated **Broader Impact** paragraph in **Appendix M**, explaining that VTA and EG can be used as offline pre-deployment auditing tools for class-wise transfer failures.
> > 3. Added **Footnote 1 in the Introduction** clarifying that “equity” refers only to class-conditional missed transfer of valid teacher knowledge, not demographic fairness or protected-group guarantees.

---

> > ### Author Response · Authors · 2026-07-19
> > **Response to Review: Part 2**
> >
> > #### **Change 5: TED objective and distinction from reweighting and DRO baselines**
> >
> > **Response:**
> > We thank the reviewer for this important feedback. TED is different from class-frequency reweighting method. It regularizes a transfer-specific residual: the amount of correct and confident teacher knowledge that is not inherited by the graph-free student. TED is formulated as a bounded class-conditional missed-transfer risk. First, for each labeled training node $v$, the frozen teacher GNN produces a probability vector $p_v^t$ and a predicted label $\hat{y}_v^t$. The student MLP produces $p_v^s$. Let $y_v$ be the ground-truth label, $V_c$ be the set of labeled training nodes from class $c$, and $q_v^t \in [0,1]$ be the normalized confidence of the teacher prediction. TED defines the reliability weight
> >
> > $w_v^t = \mathbf{1}[\hat{y}_v^t = y_v] q_v^t .$
> >
> > Thus, an incorrect teacher prediction receives zero weight, while a correct but uncertain teacher prediction is downweighted. TED then defines a teacher-targeted transfer loss
> >
> > $\ell_v^{tr}(\phi) = -\log p^s_{v,\hat{y}_v^t},$
> >
> > where $p^s_{v,\hat{y}_v^t}$ is the probability assigned by the student to the teacher’s predicted class. This loss is large when the student fails to assign high probability to the teacher decision being transferred.
> >
> > For each class $c$, TED aggregates this quantity into the class-wise missed reliable transfer risk
> >
> > $R_c(\phi)=
> > \frac{\sum_{v \in V_c} w_v^t \ell_v^{tr}(\phi)}
> > {\sum_{v \in V_c} w_v^t}.$
> >
> > $R_c(\phi)$ measures how much correct and confident teacher knowledge for class $c$ is not inherited by the graph-free student.
> >
> > This is the key distinction from class-balanced KD, focal loss, and DRO: those methods operate on label frequency, student-side example difficulty, or generic worst-class loss, whereas $R_c(\phi)$ is a teacher-conditioned transfer residual aligned with VTA.
> >
> > This distinguishes TED from the baselines in **Table 2**. Class-Balanced KD uses label frequency, Focal Loss emphasizes examples that are difficult under the student’s prediction, and DRO optimizes worst-class risk. TED instead regularizes **missed reliable transfer conditioned on teacher correctness and confidence**, which is the failure mode measured by VTA and EG.
> >
> > To make this distinction empirical, the revised manuscript adds **Balanced KD**, **Focal Loss**, and **DRO** comparisons in **Table 2**. The results show that reweighting or robust optimization does not consistently match TED’s accuracy-EG tradeoff. In the macro-average comparison, Balanced KD increases EG from **13.98% to 20.32%**, while macro-average accuracy changes from **85.86% to 85.95%**. Focal Loss increases EG slightly from **13.98% to 14.21%**, while macro-average accuracy changes from **85.86% to 86.10%**. DRO reduces EG from **13.98% to 9.99%**, but with a larger accuracy drop from **85.86% to 83.38%**. TED gives the strongest tradeoff: it reduces macro-average EG from **13.98% to 8.79%**, corresponding to a **37.12% reduction** relative to GLNN, while macro-average accuracy changes from **85.86% to 85.70%**, corresponding to only a **0.19% reduction**.
> >
> > Furthermore, we have expanded the Related Work section to discuss class-balanced, focal-style, and distributionally robust objectives more explicitly and to clarify how TED differs from these approaches.
> >
> > > **Changes in the manuscript:**
> > >
> > > 1. Clarified the TED objective as class-conditional missed reliable transfer.
> > > 2. Added **Balanced KD**, **Focal Loss**, and **DRO** comparisons in **Table 2**.
> > > 3. Revised the related-work and experimental discussion to distinguish TED from label-frequency reweighting, focal loss, and robust optimization.

---

> > > ### Author Response · Authors · 2026-07-19
> > > **Response to Review: Part 3**
> > >
> > > #### **Change 6: Worst-label agreement**
> > > **Response:**
> > > We thank the reviewer for this important suggestion. The original submission reported complementary VTA diagnostics, including **MeanVTA**, **WorstVTA**, and **StdVTA**, for three representative datasets in **Appendix J**. We also note that the original submission provided both visual and quantitative evidence for worst-label improvement. In particular, **Figures 1, 3, and 5** showed that TED augmentation improves the lowest-VTA class in representative settings.
> > >
> > > In the revised manuscript, we have therefore substantially expanded **Appendix J**. **Table 6** now reports **Accuracy**, **MeanVTA**, **WorstVTA**, **StdVTA**, and **Equity Gap** across six datasets for **GLNN**, **TED+GLNN**, and additional reweighting and robust-optimization baselines, including **Class-Balanced KD**, **Focal Loss**, and **DRO**. We have also revised the *Evaluation Metrics* paragraph in **Section 5** to explicitly state that accuracy is used as the primary predictive-utility metric, while EG is used as the primary class-conditional transfer-disparity metric and is complemented by MeanVTA, WorstVTA, StdVTA, and teacher-correct support statistics.
> > >
> > > The results in **Table 6** show that TED improves class-wise transfer reliability beyond simply reducing EG. In particular, TED improves **WorstVTA on all six datasets** and obtains the strongest average rank among the compared methods for the transfer-reliability metrics, including **WorstVTA**, **StdVTA**, and **EG**.
> > >
> > > The expanded diagnostics in Table 6 show that TED improves the worst-class transfer profile rather than merely reducing a range statistic. On CiteSeer, WorstVTA increases from **64.33% to 86.44%**, while StdVTA decreases from **11.36% to 4.07%**, and EG decreases from **33.17% to 11.41%**. On Amazon-Photo, WorstVTA increases from **76.51% to 79.37%**, while StdVTA decreases from **7.17% to 5.58%**, and EG decreases from **23.43% to 18.26%**. Similar trends hold on the remaining datasets.
> > >
> > >
> > >
> > >
> > >
> > > > **Changes in the manuscript:**
> > > >
> > > > 1. Clarified in the *Evaluation Metrics* paragraph of **Section 5** that EG is complemented by MeanVTA, WorstVTA, StdVTA, and teacher-correct support statistics.
> > > > 2. Expanded **Appendix J: Table 6** to report Accuracy, MeanVTA, WorstVTA, StdVTA, and EG across six datasets and multiple reweighting / robust-optimization baselines.
> > > > 3. Made the captions of **Figures 1, 3, and 5** explicit that TED improves the VTA of the most under-transferred / lowest-VTA class under the corresponding base GLNN method.
> > > ---
> > >
> > >
> > > #### **Change 7: Sensitivity with respect class imbalance, and heterophily**
> > >
> > > **Response:**
> > > We thank the reviewer for this suggestion. In the revised manuscript, we address this concern through targeted ablations and class-level diagnostics.
> > >
> > > First, TED explicitly accounts for teacher errors through the teacher-correctness term in the reliability weight. Incorrect teacher predictions receive zero weight and therefore do not contribute to the missed-transfer risk. We further ablate this design choice in **Figure 5** by removing teacher-correctness filtering. The results show that removing this component weakens the reduction in class-conditional transfer disparity.
> > >
> > > Second, TED accounts for teacher confidence through reliability weighting. We ablate this component in **Figure 5** by removing reliability weighting. The results show that removing reliability weighting also weakens the reduction in EG.
> > >
> > > Third, we added **RQ6: Figure 6** to diagnose class-level factors associated with low VTA. Figure 6 analyzes **class support**, **class-wise edge homophily**, and **raw-feature separability** for each class on CiteSeer and Amazon-Photo, highlighting the lowest-VTA class under vanilla GLNN. The diagnostics show that the lowest-VTA classes tend to have weaker graph-free recovery conditions, such as lower support, lower homophily, or weaker feature separability. **Figure 1** shows how VTA varies across classes and how TED improves the most under-transferred class.
> > >
> > > > **Changes in the manuscript:**
> > > >
> > > > 1. Added / clarified ablations in **Figure 5** for removing teacher-correctness filtering and reliability weighting.
> > > > 2. Added **RQ6 / Figure 6**, analyzing class support, class-wise edge homophily, and raw-feature separability for each class.
> > > > 3. Made the class-wise VTA diagnostics in **Figure 1** explicit as evidence of label-level variation in transfer behavior.
> > > > 4. Discussed confidence calibration as a potential future direction.

---

### Review · Reviewer_8fdK · 2026-07-08

**Summary Of Contributions:**

This paper studies whether graph-free MLP students inherit knowledge from GNN teachers uniformly across classes in GNN-to-MLP distillation. The authors argue that average accuracy can hide class-conditional failures: an MLP student may preserve the teacher’s behavior well on average while failing to preserve correct teacher decisions for some labels. To study this, the paper introduces Valid Transfer Agreement (VTA), a class-level metric measuring teacher-student agreement conditioned on the teacher being correct, and Equity Gap, the range of class-wise VTA values. The paper then proposes TED, a training-time regularizer that penalizes labels with excessive missed reliable transfer, where teacher reliability is defined using both correctness and confidence. The method is designed as an augmentation to existing graph-free distillation methods and does not change the student architecture or inference-time procedure. The experiments evaluate TED across homophilic and heterophilic node-classification datasets, multiple base distillation methods, and different teacher architectures.

**Audience:**

Yes

**Audience Explanation:**

The paper identifies a meaningful and underexplored reliability issue in graph-free GNN distillation: class-wise transfer behavior can differ substantially even when average accuracy is competitive.

**Claims And Evidence:**

No

**Claims Explanation:**

The paper provides convincing evidence for the more specific claim that class-wise disparities exist in teacher–student transfer and that TED can reduce the proposed Equity Gap metric while preserving comparable average accuracy. Table 1 shows consistent reductions in Equity Gap across multiple homophilic datasets, base distillation methods, and both transductive and inductive settings, with only small changes in accuracy. The additional GCN-teacher results in the appendix further suggest that the observed effect is not tied to a particular teacher architecture. The heterophilic experiments also indicate that this issue is not limited to homophilic graphs.

That said, the evidence is less complete for the paper’s broader framing around “transfer equity” and “uniform inheritance.” The main metric, Equity Gap, is defined only as the difference between the maximum and minimum VTA across classes. While this is a useful starting point, it is incomplete on its own. It does not show whether TED consistently improves the worst-performing class, how the average VTA changes, or whether the gap sometimes shrinks because VTA decreases on easier classes. The results would be more informative if they also reported Worst VTA, Mean VTA, Std VTA, and the number of teacher-correct samples per class. This is particularly important because VTA can become undefined or unstable when the teacher-correct support is small, as illustrated by the Cornell discussion, where one class has no evaluable transfer support.

The method itself is technically reasonable, but its optimization objective relies on teacher correctness over labeled training nodes. This is appropriate for the supervised or semi-supervised setting considered in the paper, but the paper should make clearer that TED’s reliability signal depends on the availability of labeled data and may become noisy when each class has only a small number of labeled examples. Although the conclusion mentions this as a limitation, it is central enough to deserve empirical analysis, for example by varying the number of labeled nodes per class.

Overall, I think the core claims are mostly supported when interpreted within the proposed VTA and Equity Gap evaluation framework. However, the paper should either provide stronger evidence for its broader claims about uniform knowledge inheritance or soften those claims so that they more closely match what the current experiments directly demonstrate.

**Requested Changes:**

1. **Report more complete transfer metrics beyond Equity Gap.**
    Please include Worst VTA, Mean VTA, Std VTA, and, if possible, class-wise teacher-correct support sizes in the main results or in a prominent table. Equity Gap alone is not sufficient because it is a range statistic: it does not show whether TED improves the worst-transferred class or simply compresses the range across classes.
2. **Clarify the stability of VTA and Equity Gap under small class support.**
    Since VTA is computed only over test samples that the teacher predicts correctly, some classes may have very small denominators. The paper should report the number of valid teacher-correct samples per class, or at least the minimum and median support across datasets. This is especially important for datasets such as Cornell and Wisconsin, where small support sizes may make VTA and Equity Gap unstable.
3. **Clarify hyperparameter selection and validation criteria.**
    The paper states that TED-specific parameters are tuned on the validation split, but it is not fully clear whether model selection is based on accuracy, Equity Gap, a combined objective, or another criterion. Since TED directly targets equity-related metrics, the validation protocol should be specified carefully to ensure a fair comparison with the baselines.
4. **Strengthen or qualify the claims about “equity.”**
    The paper appropriately notes that its notion of equity is not demographic fairness. However, the term “equity” may still suggest a stronger normative claim than the experiments establish. The authors could clarify that the paper studies class-conditional transfer reliability rather than fairness over sensitive groups, and avoid implying broader practical fairness conclusions that are not directly supported by the experiments.
5. **Discuss the dependence on labeled teacher correctness more deeply.**
    TED defines reliable teacher knowledge using ground-truth correctness on labeled nodes. This is reasonable for the proposed setting, but it may limit the method in low-label or weakly supervised scenarios. It's better to either add an experiment varying the amount of labeled data or more explicitly present this dependence as a central limitation.

---

> ### Author Response · Authors · 2026-07-19
> **Response to Review: Part 1**
>
> We sincerely thank the reviewer for the constructive feedback, and for recognizing the relevance of this problem to the TMLR audience. We are encouraged that the reviewer acknowledges the importance of identifying class-conditional reliability issues in graph-free GNN-to-MLP distillation, where competitive average accuracy can obscure substantial disparities in how valid teacher knowledge is inherited across classes.
>
>
> Below, we provide detailed point-by-point responses to the reviewer’s requested changes. All corresponding revisions are highlighted in yellow in the updated manuscript.
>
> ---
>
> #### **Change 1: Transfer metrics beyond Equity Gap**
>
> **Response:**
> We thank the reviewer for this important suggestion. The original submission reported complementary VTA diagnostics, including **MeanVTA**, **WorstVTA**, and **StdVTA**, for three representative datasets in **Appendix J**. We also note that the original submission provided both visual and quantitative evidence for worst-label improvement. In particular, **Figures 1, 3, and 5** showed that TED augmentation improves the lowest-VTA class in representative settings.
>
> In the revised manuscript, we have therefore substantially expanded **Appendix J**. **Table 6** now reports **Accuracy**, **MeanVTA**, **WorstVTA**, **StdVTA**, and **Equity Gap** across six datasets for **GLNN**, **TED+GLNN**, and additional reweighting and robust-optimization baselines, including **Class-Balanced KD**, **Focal Loss**, and **DRO**. We have also revised the *Evaluation Metrics* paragraph in **Section 5** to explicitly state that accuracy is used as the primary predictive-utility metric, while EG is used as the primary class-conditional transfer-disparity metric and is complemented by MeanVTA, WorstVTA, StdVTA, and teacher-correct support statistics.
>
> The results in **Table 6** show that TED improves class-wise transfer reliability beyond simply reducing EG. In particular, TED improves **WorstVTA on all six datasets** and obtains the strongest average rank among the compared methods for the transfer-reliability metrics, including **WorstVTA**, **StdVTA**, and **EG**.
>
> The expanded diagnostics in Table 6 show that TED improves the worst-class transfer profile rather than merely reducing a range statistic. On CiteSeer, WorstVTA increases from **64.33% to 86.44%**, while StdVTA decreases from **11.36% to 4.07%**, and EG decreases from **33.17% to 11.41%**. On Amazon-Photo, WorstVTA increases from **76.51% to 79.37%**, while StdVTA decreases from **7.17% to 5.58%**, and EG decreases from **23.43% to 18.26%**. Similar trends hold on the remaining datasets.
>
>
>
>
>
> > **Changes in the manuscript:**
> >
> > 1. Clarified in the *Evaluation Metrics* paragraph of **Section 5** that EG is complemented by MeanVTA, WorstVTA, StdVTA, and teacher-correct support statistics.
> > 2. Expanded **Appendix J: Table 6** to report Accuracy, MeanVTA, WorstVTA, StdVTA, and EG across six datasets and multiple reweighting / robust-optimization baselines.
> > 3. Made the captions of **Figures 1, 3, and 5** explicit that TED improves the VTA of the most under-transferred / lowest-VTA class under the corresponding base GLNN method.
>
> ---
>
> #### **Comment 2: Teacher-correct support size**
>
> **Response:**
> We thank the reviewer for this important suggestion. In the revised manuscript, we added teacher-correct support statistics in **Appendix J: Table 7**. Table 7 reports the total number of teacher-correct evaluation samples, the minimum support across classes, the median support across classes where defined, and the full class-wise teacher-correct support. We report these statistics for the **GraphSAGE teacher** on the six homophilic datasets and for the **ACM-GCN teacher** on Cornell and Wisconsin.
>
>
> > **Changes in the manuscript:**
> >
> > 1. Added **Appendix J: Table 7**, reporting total, minimum, median where defined, and complete class-wise teacher-correct support.

---

> > ### Author Response · Authors · 2026-07-19
> > **Response to Review: Part 2**
> >
> > #### **Change 3: Dependence on labeled nodes**
> >
> > **Response:**
> > We thank the reviewer for raising this central limitation. The original submission already acknowledged in the conclusion that TED relies on labeled nodes to identify teacher-correct decisions, which may be restrictive in scarcely labeled settings.
> >
> > In the revised manuscript, we added a dedicated **Appendix M** on low-label sensitivity and limitations. This section explicitly discusses the dependence of TED’s reliability signal on labeled teacher-correct support and evaluates TED under extremely scarce supervision using **1-shot** and **5-shot** node classification on CiteSeer and Amazon-Photo, where \(k\)-shot denotes \(k\) labeled training nodes per class.
> >
> > In the extreme **1-shot** setting, TED still reduces EG on both datasets, although the improvement is modest because the correctness-based reliability signal is estimated from only one labeled example per class. On CiteSeer, EG decreases from **98.50% to 95.36%**, with accuracy remaining at **28.12%**. On Amazon-Photo, EG decreases from **62.78% to 60.06%**, while accuracy changes from **75.88% to 72.78%**.
> >
> > The benefit becomes clearer in the **5-shot** setting. On CiteSeer, TED improves accuracy from **59.80% to 60.73%** while reducing EG from **87.14% to 58.41%**. On Amazon-Photo, TED reduces EG from **42.40% to 37.47%**, with a comparatively small accuracy change from **88.03% to 87.48%**.
> >
> > We have also expanded the limitations to state more clearly that TED is most appropriate for supervised or semi-supervised graph-free distillation settings with nontrivial labeled support, and that reducing dependence on labeled teacher correctness is an important direction for future work.
> >
> > > **Changes in the manuscript:**
> > >
> > > 1. Added **Appendix M: Limitations and Future Work**.
> > > 2. Added **1-shot** and **5-shot** low-label sensitivity experiments on CiteSeer and Amazon-Photo.
> > > 3. Expanded the limitations discussion to emphasize the dependence on labeled teacher-correct support.
> >
> > ---
> >
> > #### **Change 4: Hyperparameter selection**
> >
> > **Response:**
> > We thank the reviewer for the important suggestion. In the revised manuscript, we clarify this in **Appendix F**.
> >
> > For all methods, including TED-augmented variants, hyperparameters are selected using the validation split only. We use **validation accuracy** as the hyperparameter-selection criterion.
> >
> > > **Changes in the manuscript:**
> > >
> > > 1. Clarified the hyperparameter-selection protocol in **Appendix F**.
> > > 2. Stated explicitly that validation accuracy is used for model selection across methods.
> >
> >
> > ---
> >
> > #### **Change 5: Clarification regarding “Equity”**
> >
> > **Response:**
> > We thank the reviewer for this thoughtful observation. The original manuscript already distinguished our use of “equity” from demographic fairness in the Introduction. Specifically, our concern is not fairness over sensitive attributes, but rather a deployment reliability issue: a graph-free student may systematically fail to inherit valid GNN teacher knowledge for some class labels.
> >
> > Following the reviewer’s suggestion, we have further tightened this terminology throughout the revised manuscript. We now consistently describe the contribution as addressing **class-conditional transfer reliability** and **disparity in valid teacher-knowledge inheritance**. We also added an explicit **footnote** in the **Section 1: Introduction** clarifying that “equity” in TED refers only to label-wise missed transfer of valid teacher knowledge in graph-free GNN-to-MLP distillation, and should not be interpreted as demographic fairness or a protected-group fairness guarantee.
> >
> > > **Changes in the manuscript:**
> > >
> > > 1. Added **Footnote 1** in the Introduction to explicitly clarify the intended meaning of “equity.”
> > > 2. Clarified that TED does not make claims about demographic fairness, sensitive attributes, or protected-group guarantees.
> > > 3. Revised the surrounding language to emphasize class-conditional transfer reliability and deployment reliability.

---

### Author Response · Authors · 2026-07-19
**Summary of revisions addressing reviewer suggestions**

We thank all reviewers for their constructive feedback. The main changes in the manuscript are summarized below.

| Reviewer concern | Original submission | Revised manuscript |
|---|---|---|
| **Transfer metrics beyond Equity Gap** | The original submission complemented EG with **MeanVTA**, **WorstVTA**, and **StdVTA** in **Appendix J**, reported on three representative datasets for GLNN and TED. | We have made this more explicit and prominent. In **Section 5: Evaluation Metrics**, we now state clearly that EG is complemented by **MeanVTA**, **WorstVTA**, and **StdVTA**. We also expanded **Appendix J: Table 6** from three representative datasets to six datasets, and now compare GLNN, Balanced KD, DRO, Focal Loss, and TED using **Accuracy**, **MeanVTA**, **WorstVTA**, **StdVTA**, and **Equity Gap**. |
| **Evidence that TED improves worst-label transfer** | The original submission provided visual and quantitative evidence for worst-label improvement. **Figures 1, 3, and 5** showed that TED augmentation improves the lowest-VTA class in representative settings. In addition, **Appendix J** reported **WorstVTA**, **MeanVTA**, **StdVTA**, and EG on three representative datasets. | In the captions of **Figures 1, 3, and 5**, we now state directly that TED improves the VTA of the most under-transferred class under the base GLNN method. We also expanded **Appendix J: Table 6** from three representative datasets to six datasets, and now compare GLNN, Balanced KD, DRO, Focal Loss, and TED using **Accuracy**, **MeanVTA**, **WorstVTA**, **StdVTA**, and **Equity Gap**. |
| **Stronger reweighting and robust-optimization baselines** | The original submission primarily compared TED with graph-free distillation methods such as GLNN, KRD, and FAITH. To test whether a balancing mechanism could explain the gains, the original manuscript also included a representative **class-balanced GLNN** baseline in **Figure 5** under **RQ4**. | We have substantially strengthened this comparison. In the revised manuscript, we added **Focal Loss** and **Distributionally Robust Optimization (DRO)** baselines in addition to the existing **class-balanced** baseline, and introduced a dedicated experimental question (**RQ2**). The expanded results are reported in **Table 2**. We also revised **Section 2** to clarify the distinction between TED and these baselines: class-balanced KD uses label frequency, focal loss emphasizes student-side difficult examples, and DRO optimizes worst-class risk, whereas TED regularizes **missed reliable transfer conditioned on teacher correctness and confidence**, which is the specific failure mode measured by VTA and EG. |
| **Predictive metrics beyond average accuracy** | Accuracy was the primary predictive-utility metric in the main results. | We now additionally report **macro-F1** and **minority-class recall** in **Appendix K: Table 8**. |
| **Ablations for TED components** | The original submission included TED **without reliability weighting** in **Figure 5**, which tested whether the confidence-based part of the reliability signal is necessary. | We now add the complementary ablation that removes **teacher-correctness filtering**, also in **Figure 5**. Together, these ablations disentangle TED’s reliability design. |
| **VTA support** | Support sizes were not prominently reported. | We now report teacher-correct support sizes in **Table 7**. |
| **Use of “Equity”** | As noted by the reviewer, the original submission already scoped this term carefully in the **Introduction**: we stated that our concern is distinct from demographic fairness and that class labels are not treated as sensitive attributes. | We have made this scope even more explicit and consistent throughout the revised manuscript. We now describe the contribution as addressing **class-conditional transfer reliability** and **disparity in valid teacher-knowledge inheritance**, rather than fairness over sensitive groups. We also added an explicit footnote in the **Introduction** clarifying that “equity” in TED refers only to label-wise missed transfer of valid teacher knowledge, not to demographic fairness or protected-group guarantees. |
| **Dependence on labeled teacher-correct samples** | As noted by the reviewer, the original submission already identified this dependence as a limitation. | We have now made this limitation more central and empirically examined it. **Appendix M** adds a **low-label sensitivity analysis**, including extremely scarce-label **1-shot** and **5-shot** node classification settings in **Figure 13**. |
|**Mechanistic explanation**| The class-level factors associated with low VTA were not analyzed explicitly. | We added a new diagnostic analysis under **RQ6: Diagnosing Class-Wise Transfer Disparities**. In **Figure 6**, we analyze three class-level factors: **class support**, **class-wise edge homophily**, and **raw-feature separability**.|